evolution, behaviour

inbreeding avoidance, inbreeding depression, random mating

**Author for correspondence:**
Victoria L. Pike
e-mail: victoria.pike@zoo.ox.ac.uk

†These authors contributed equally to this study.

# Why don't all animals avoid inbreeding?

Victoria L. Pike[1], Charlie K. Cornwallis[2,†] and Ashleigh S. Griffin[1,†]

[1]Department of Zoology, University of Oxford, Oxford OX1 3SZ, UK
[2]Department of Biology, Lund University, Lund, Sweden

VLP, 0000-0002-4642-3506; CKC, 0000-0003-1308-3995; ASG, 0000-0001-7674-9825

Individuals are expected to avoid mating with relatives as inbreeding can reduce offspring fitness, a phenomenon known as inbreeding depression. This has led to the widespread assumption that selection will favour individuals that avoid mating with relatives. However, the strength of inbreeding avoidance is variable across species and there are numerous cases where related mates are not avoided. Here we test if the frequency that related males and females encounter each other explains variation in inbreeding avoidance using phylogenetic meta-analysis of 41 different species from six classes across the animal kingdom. In species reported to mate randomly with respect to relatedness, individuals were either unlikely to encounter relatives, or inbreeding had negligible effects on offspring fitness. Mechanisms for avoiding inbreeding, including active mate choice, post-copulatory processes and sex-biased dispersal, were only found in species with inbreeding depression. These results help explain why some species seem to care more about inbreeding than others: inbreeding avoidance through mate choice only evolves when there is both a risk of inbreeding depression and related sexual partners frequently encounter each other.

## 1. Introduction

Mating with relatives (inbreeding) can result in offspring with reduced survival and fertility, a phenomenon known as inbreeding depression [1]. Inbreeding depression has been observed to occur in wild animals with severe consequences [2,3]. This has led to the prediction that selection will favour individuals that base mate choice decisions on relatedness [2,4–7]. However, studies of mate choice have failed to provide consistent support for this prediction, with a recent meta-analysis highlighting the lack of evidence for inbreeding avoidance across species [8]. While many species have been found to strongly prefer unrelated mates, for example, long-tailed tits, *Aegithalos caudatus* [9], in other species, such as yellow-bellied toads *Bombina variegata* [10] and common terns *Sterna hirundo* [11], mate choice decisions are not influenced by relatedness. Some species even show a preference for mating with relatives, such as the ground tit *Parus humilis* [12], the cichlid fish *Pelvicachromis taeniatus* [13,14] and White's skink *Liopholis whitii* [15]. Why do some species actively avoid mating with relatives while others do not, and does this pose a problem for our understanding of inbreeding depression and mate choice?

A potential answer to why species vary in the extent to which they choose unrelated mates may be found by considering how inbreeding is avoided [4,5,16–18]. Individuals can reduce the chance of mating with relatives via a number of different mechanisms, including sex-biased dispersal, kin recognition and extra-pair copulations [4,5,19]. Some mechanisms, such as sex-biased dispersal, are expected to reduce average relatedness in the population, resulting in a low risk of inbreeding and weakening selection for mate avoidance [17]. In olive baboons *Papio anubis*, for example, relatedness between males and females is low because of sex-biased dispersal [20], a phenomenon that is widespread across mammals and birds [18]. In other species, sex-biased dispersal is absent and avoiding inbreeding may require other mechanisms, such as use of genetic cues for distinguishing between related and unrelated individuals [9,21].

Proc. R. Soc. B **288**: 20211045

The mechanism of inbreeding avoidance determines the strength of selection for actively discriminating against related mates through two key conditions. First, there must be a chance of encountering a related sexual partner (i.e. the risk of inbreeding is high) [22]. If related mates do not interact, then selection for active mate avoidance will be weak or absent, even if there is inbreeding depression [16]. Second, choosing unrelated mates is only predicted to occur when there is a risk of inbreeding depression [23]. The severity of inbreeding depression can be highly variable both within [2] and across species, with there being strong reductions in fitness sometimes, such as in collared flycatchers *Ficedula albicollis* [24], while in other species inbreeding can even be beneficial [13,25,26]. It is only when these two conditions are combined—related individuals interact and there is inbreeding depression—that there will be selection for mate avoidance. However, the risks of mating with relatives are often overlooked in studies of inbreeding avoidance and are rarely matched with data on inbreeding depression.

To understand variation in inbreeding avoidance across species, we conducted a phylogenetic meta-analysis collecting data on relatedness between mates, average male-female relatedness within populations, inbreeding depression and mechanisms of inbreeding avoidance that do not require active mate choice (mating behaviour and sex-biased dispersal). Data were extracted from 40 studies on 41 species from across the animal kingdom. The presence and absence of inbreeding depression were recorded based on reports from authors, as well as calculating an effect size of the relationship between offspring fitness and the level of inbreeding, which was possible for a subset of species (16 species). Using these data, we addressed the following questions. Is variation in relatedness between mates explained by the frequency that relatives encounter each other and inbreeding depression? Are all mechanisms of inbreeding avoidance equally effective? What mechanism of inbreeding avoidance do species use most frequently? Is random mating with respect to relatedness more likely to occur when other mechanisms of inbreeding avoidance reduce the frequency with which relatives interact?

## 2. Methods

### (a) Literature search
We conducted a literature search on Web of Science using the keyword search strings 'inbreeding avoidance', 'incest avoidance', 'inbreeding preference' and 'incest preference' up to and including 6 January 2021 (see electronic supplementary material, figure S1 for PRISMA flow chart [27] and electronic supplementary material, tables S24–S26 for a full list of studies and searches). We manually screened these results based on titles and abstracts. Studies on humans, plants and captive populations were excluded, which made up a large proportion of the search results (electronic supplementary material, figure S1) as they are often used to investigate inbreeding [2,28]. Next, we performed full text screening of articles of interest (for details of the inclusion and exclusion, see electronic supplementary material, table S24) and carried out backwards and forwards citation searches on the studies. Our final dataset comprised 40 papers representing 41 different species from six different classes: Aves, Mammalia, Reptilia, Amphibia, Insecta and Actinopterygii.

### (b) Data collection
#### (i) Estimating inbreeding avoidance
The standard measure of inbreeding is Wright's inbreeding coefficient ($f$) [29]. The value of $f$ represents the probability that two alleles in an individual are identical by descent [1,29]. However, $f$ does not indicate the level of inbreeding avoidance, as it does not consider levels of relatedness across the whole population. We therefore analysed the strength of inbreeding avoidance by examining the degree of relatedness between pairs in relation to the average relatedness between males and females in the study population (table 1). To do this, we collected two measures from each study population:

(i) The average relatedness between mating pairs (*rPairs*).
(ii) The average relatedness between males and females in the population (*rAverage*).

For two species (*Pan troglodytes* and *Marmota flaviventris*) there were estimates from two different populations with the same sample size for which averages of *rPairs* and *rAverage* were included in analyses. Consequently, all analyses included one estimate per species. Author reports of whether mate choice was random with respect to relatedness (the presence and absence of inbreeding avoidance) and estimates of population size were also recorded for each study (table 1).

Estimates of relatedness between mated pairs and opposite sex pairs are correlation coefficients bounded by −1 (lower than average relatedness) and 1 (clonal population) [30]. These estimates were transformed using Fisher's Z ($Zr$) before analysis using the following formula:

$$Z_r = \frac{1}{2}\log_e\left|\frac{1+r}{1-r}\right|.$$

Throughout the analyses, these are referred to as: (i) *ZrPairs*, transformed effect sizes of mean relatedness between breeding pairs in the study population; and (ii) *ZrAverage*, transformed effect sizes of mean relatedness between all pairs in the study population which we consider to be the pool of potential mates.

#### (ii) Classifying mechanisms of inbreeding avoidance
We classified the mechanisms of inbreeding avoidance according to the author's evidence. We classified the mechanisms into three groups: active mate choice, sex-biased dispersal and post-mating avoidance (table 1). A species was classified as having 'active mate choice' when the author reported evidence of any type of mate choice based on relatedness. When the author had reported evidence of female- or male-biased dispersal from the natal group prior to mating, a species was classified as having sex-biased dispersal. Post-mating avoidance was defined as any mechanism of inbreeding avoidance occurring after mating including extra-pair copulations (in these cases extra-pair copulations were a mechanism of avoiding inbreeding, although this is not always the case [31]) and post-copulatory inbreeding avoidance. Species without any known mechanism of inbreeding avoidance were classified as 'none' (table 1). We also recorded whether a species was a cooperative breeder, as they frequently interact with relatives which could facilitate the recognition of relatives.

#### (iii) Estimating inbreeding depression
Inbreeding depression was estimated in two ways. First, we calculated an effect size of inbreeding depression as the Pearson's correlation coefficient ($r$) between fitness and levels of inbreeding (*rDepression*). Studies presented data on inbreeding depression as means, standard deviations and the sample sizes of inbred versus outbred offspring that were converted to $r$. Estimates of *rDepression* were transformed to Fisher's Zr using the formula

**Table 1.** Details of data collection.

| information required | data collected |
|---|---|
| does the species avoid inbreeding? | authors' statement on inbreeding avoidance (yes/no) |
| | if the author did not find any evidence for inbreeding avoidance, we classified the species as having no mechanism of inbreeding avoidance in our analysis |
| what method of inbreeding avoidance does a species employ? | mechanism(s) of inbreeding avoidance recorded by the author: |
| | active mate choice (yes/no)? |
| | sex-biased dispersal (yes/no)? |
| | post-mating avoidance (post-copulatory and extra-pair copulation) (yes/no)? |
| does the species have inbreeding depression? | authors' statement on inbreeding depression (yes/no) |
| | if this was not available and no other studies investigated inbreeding in the species, risk of inbreeding depression was assumed to exist |
| | quantitative estimates of inbreeding depression were included in analysis if available (a mean value or using a correlation between inbreeding and fitness) |
| relatedness between pairs (*rPairs*) | the average relatedness reported between breeding pairs in the study population |
| average relatedness between males and females in the population (*rAverage*) | the average relatedness between males and females in the study population. We considered these pairs to be 'potential mates' |
| | we excluded studies where the relatedness between males and females in the population may be affected by other factors, e.g. if all opposite sex pairs in a population are related |
| information on species dispersal | if all individuals disperse |
| | if dispersal is sex-biased |
| | if mating occurs pre- or post-dispersal |
| does the species mate randomly? | species were considered to mate randomly if there was no effect of relatedness on mate choice |

above (*ZrDepression*). Values were extracted from the studies in our main analysis, except for three cases [32–34] where different studies were cited [35–37]. Values were either extracted from the main text of the study or from figures using 'Webplotdigitizer' [38]. Inbreeding depression was estimated using a variety of different fitness measures [3,39,40] which we categorized into three groups; reproductive success ($N_{Species} = 5$), mortality ($N_{Species} = 7$) and body mass ($N_{Species} = 4$). We examined the sensitivity of our results to the type of fitness measure used by analysing differences in *ZrDepression* between these categories (see electronic supplementary material).

Inbreeding depression is often difficult to quantify in natural populations [3,26]. For example, in some species, the occurrence of inbreeding is too rare to calculate any associated costs (e.g. [9]). As a result, it was only possible to get estimates of *ZrDepression* for 16 species. In order to examine inbreeding depression in a greater number of species, a second method based on authors statements on the presence or the absence of inbreeding depression was used to classify species (referred to as 'reported inbreeding depression'; table 1). Where authors did not report whether a species suffered from inbreeding depression or not (not mentioned) we assumed species suffer deleterious consequences of mating with related individuals, as this is the most likely outcome [41] (for analyses and discussion of this assumption see electronic supplementary material). For details of classifications for all species, see electronic supplementary material, table S1. We examined the robustness of reported inbreeding depression by examining the relationship between reported measures of inbreeding depression and *ZrDepression* (see electronic supplementary material). There was good correspondence between the methods, and we therefore present analyses using reported inbreeding depression due to the larger sample size (41 species versus 16 species). Analyses using

*ZrDepression* are presented in the supplementary information (electronic supplementary material, tables S14–20 and S22).

## (c) Statistical analysis
### (i) General statistical techniques
All statistical analysis were conducted in R [42]. Data were analysed using Bayesian phylogenetic multi-level meta-regressions (BPMM) with Markov chain Monte Carlo (MCMC) estimation implemented in the 'MCMCglmm' package [43]. Data were from a taxonomically diverse range of species. Phylogenetic non-independence between species was accounted for by creating a phylogenetic tree (figure 1*a*) using the R package 'rotl' [44] that accesses information from the open tree of life [45]. A phylogenetic variance–covariance matrix was included in all models as a random effect. Data points were weighted by the inverse sampling variance $(1/(n-3))$ associated with each of the effect sizes *ZrPairs*, *ZrAverage* and *ZrDepression* using the 'mev' term in MCMCglmm.

### (ii) Specific analyses
We ran seven models in total. First, intercept-only BPMMs were used to quantify heterogeneity, $I^2$ [46,47] in *ZrPairs*, *ZrAverage* and *ZrDepression* (models 1.1, 1.4 and 1.7). We calculated phylogenetic heritability ($H^2$) and total heterogeneity ($I^2_{Total}$) as specified in [47] using modified code from [48]) which was as follows:

$$H^2 = \frac{\sigma^2_{phylogeny}}{\sigma^2_{phylogeny} + \sigma^2_{residual}}$$

and

$$I^2_{Total} = \frac{\sigma^2_{phylogeny} + \sigma^2_{residual}}{\sigma^2_{phylogeny} + \sigma^2_{residual} + \sigma^2_m},$$

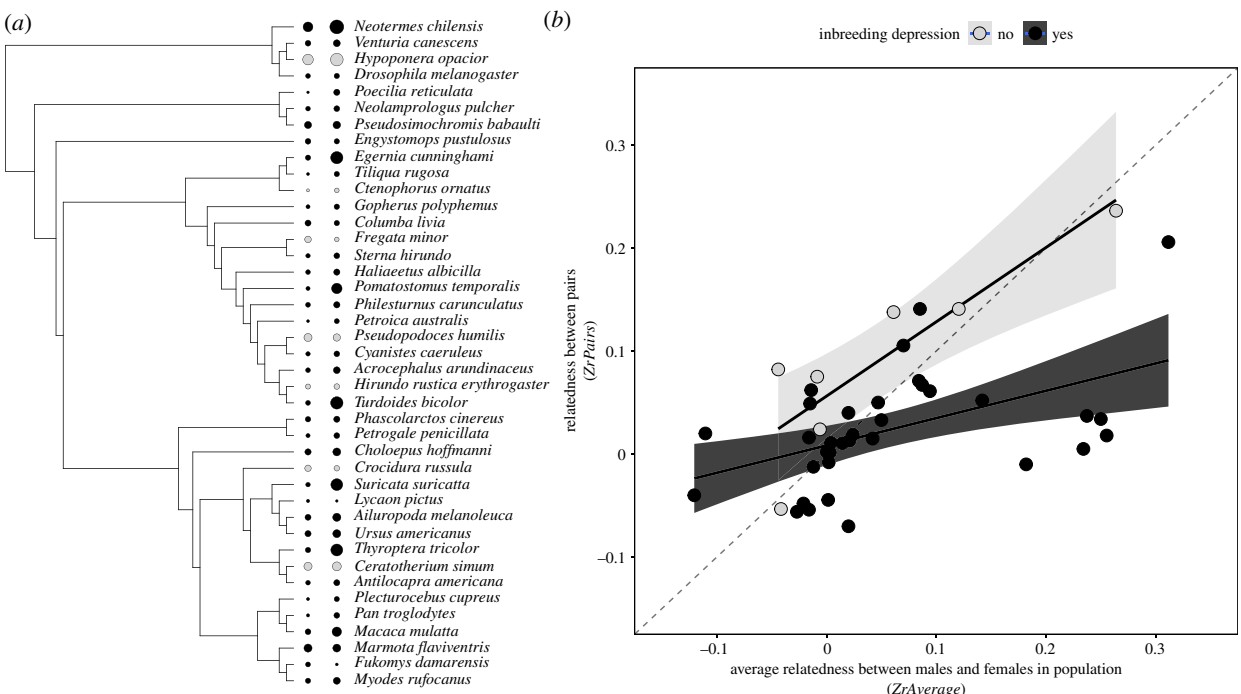

**Figure 1.** Variation in inbreeding avoidance across species. (*a*) Phylogenetic tree of species included in the analysis with *ZrPairs* (left) and *ZrAverage* (right) represented by the size of points and inbreeding depression by colour (black = reported inbreeding depression, grey = no reported inbreeding depression). (*b*) The relationship between average relatedness between males and females in the population and between mates for species with (black) and without (grey) reported inbreeding depression. The dashed line represents 1 : 1 relationship and the solid lines and shaded areas are regression lines with 95% confidence intervals.

where $\sigma_m^2$ is the measurement error variance. Second, we tested if there was a general effect of inbreeding avoidance across species using a BPMM with *ZrPairs* as the response variable, and *ZrAverage* as a fixed effect (Model 2). If there is no evidence for inbreeding avoidance across species then there will be a 1 : 1 relationship between *ZrPairs* and *ZrAverage,* indicating no deviation in relatedness between mated pairs from between random pairs of males and females from the population. This was tested by examining if the 95% credible interval (CI) of the slope of *ZrAverage* encompassed one. A slope of *ZrAverage* statistically significantly less than 1 is expected if there is inbreeding avoidance, indicating lower relatedness between mated pairs than the average relatedness between random pairs from the population. Third, the impact of inbreeding depression on mate choice for unrelated mates was tested using a BPMM with *ZrPairs* as the response variable, and *ZrAverage,* reported inbreeding depression (categorical variable with two levels: yes/no) and their interaction term as the fixed effects (Model 3). Fitting the interaction between inbreeding depression and *ZrAverage* enabled us to test if there was inbreeding avoidance (relationship between *ZrPairs* and *ZrAverage* significantly less than one) for species with and without inbreeding depression.

Fourth, we tested if the probability of evolving at least one mechanism for avoiding inbreeding was dependent on inbreeding depression using a BPMM (binomial error distribution with logit link function). The response variable was the number pairs with a mechanism of inbreeding avoidance versus number pairs without. In practice, authors reported that species either had or did not have a mechanism of inbreeding avoidance, but using number of pairs as a binomial response rather than a binary response of presence absence enabled differences in sample size across studies to be modelled. Reported inbreeding depression was fitted as a fixed effect. This model (Model 4) and the following three (Models 5–8) were restricted to only species with reported evidence of inbreeding depression ($N_{\text{species}} = 34$). Model 5 investigated whether different mechanisms of inbreeding avoidance were equally effective at reducing

relatedness between pairs, and more effective than the species without a known mechanism for inbreeding avoidance. *ZrPairs* was the response variable and the mechanism of inbreeding avoidance (four-level categorical variable; active mate choice, sex-bias dispersal, post-mating avoidance and no avoidance), *ZrAverage* and their interaction were fitted as fixed effects. Model 6 investigated whether some mechanisms of inbreeding avoidance work by reducing the probability of individuals encountering related mates. *ZrAverage* was the response variable, and the mechanism of inbreeding avoidance was a fixed effect. Model 7 tested whether choosing mates randomly with respect to relatedness was more likely to occur when other mechanisms of inbreeding avoidance reduced average relatedness in populations. Random mating was a binomial response variable (logit link function) of the number of pairs with and without random mating and the mechanism of inbreeding avoidance was included as a fixed effect. Model 8 addressed whether there was an effect of cooperative breeding on inbreeding avoidance. *ZrPairs* was the response variable with cooperative breeding (two-level categorical variable: yes, no), *ZrAverage* and their interaction term as fixed effects. A social insect was excluded from this analysis as it did not fit into the category of cooperative or non-cooperative breeder leaving 33 species.

Default priors were used for fixed effects (independent normal priors with zero mean and large variance ($10^{10}$)) and inverse gamma priors were used ($V = 1$, nu = 0.002) were used for random effects. Gaussian models were run for 2 000 000 iterations with a burn-in of 1 000 000 and a thinning level of 1000 and binomial models were run for 6 000 000 iterations with a burn-in of 1 000 000 and a thinning level of 5000. We checked for convergence of the models by running them three times, and visually checking their traces. Convergence was quantitively assessed using Gelman and Rubin's convergence diagnostic (see electronic supplementary material, tables S21 and S23). We selected the results of one of the three models at random for estimating model parameters. Results were considered to be

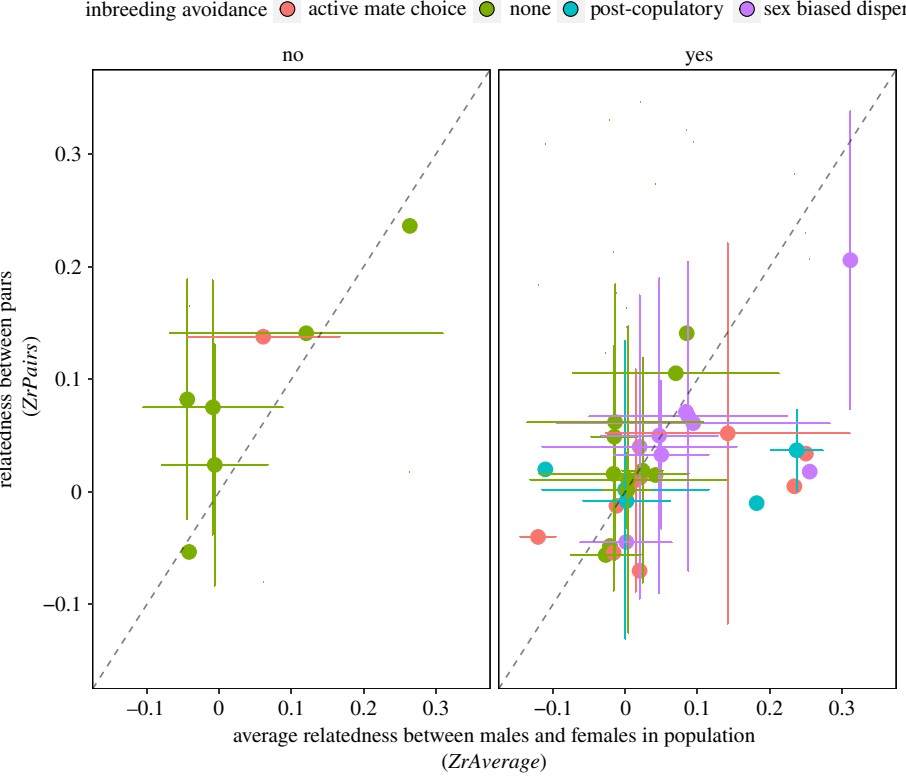

**Figure 2.** Inbreeding avoidance depends on how frequently relatives are encountered and the presence of inbreeding depression. In species with inbreeding depression, relatedness between mates is consistently low including in populations where average relatedness between potential partners is high whereas for species without inbreeding depression related partners frequently encounter each other and there is no active avoidance of related mates. (a) Species without reported inbreeding depression and (b) shows species with inbreeding depression. Points represent mean values ± standard error. The dashed line represents the 1 : 1 relationship. (Online version in colour.)

statistically significant when the 95% CI's for differences did not overlap specified values (e.g. slope of 1) and when the pMCMC (percentage of iterations above or below a test value correcting for the finite sample size of posterior samples) values were less than 0.05. The percentage of variation explained by fixed effects ($R^2_{\mathrm{marginal}}$) was calculated for all BPMMs following [49] using code modified from [49].

### (iii) Publication bias tests

Evidence for publication bias was examined in three ways for *ZrPairs* and *ZrDepression*. First, asymmetry in funnel plots was examined using meta-analytic residuals from intercept-only models plotted against precision estimates (square root of inverse sampling variance). Second, an Egger's regression test for funnel plot asymmetry was performed by fitting models with the effect size as the response variable weighted by their inverse sampling variances, the standard error of effect size as a fixed effect and the same random effects used in meta-analytical models. For *ZrPairs*, we also included *ZrAverage* as we expect publication bias to influence the residual variation around the relationship between *ZrPairs* and *ZrAverage*: high values of *ZrPairs* relative to *ZrAverage* are used to conclude there is inbreeding avoidance. There was no evidence of publication bias for *ZrPairs* or *ZrDepression* (see Models PB1–PB3; electronic supplementary material, figures S2–S3, tables S10 and S12). Third, a test for time-lag bias was conducted by fitting the same models as specified for the Egger's regressions, but the 'year of publication' was included as a fixed effect instead of the effect size standard error (as in [50]). There was no evidence of time-lag bias (CIs of year spanned 0 in all models: electronic supplementary material, tables S11 and S13).

For all data, code, model runs, results of analyses (electronic supplementary material, tables S1–S20) and methods and results

of the verification analyses see electronic supplementary material. All details of the R packages used and their versions can be found in the supplementary code (session_info object).

## 3. Results

### (a) Animals generally avoid inbreeding

We found that species generally avoid inbreeding (figure 1). Average relatedness between pairs was close to zero and significantly lower than expected from population averages (slope of *ZrAverage* versus 1: posterior mean (PM) = −0.8, CI = −1.25 to −0.42, pMCMC < 0.0001, electronic supplementary material, tables S2 and S3). There were exceptions, such as the social ant *Hypoponera opacior* [51] that had relatively high rates of inbreeding, but overall relatedness between pairs was consistently low (figure 1). Consequently, there was relatively little variation in estimates of relatedness between pairs ($I^2_{\mathrm{Total}} = 21\%$, CI = 5 to 42%, electronic supplementary material, table S2), particularly compared to variation in average relatedness in populations and inbreeding depression (*ZrAverage*: $I^2_{\mathrm{Total}} = 37\%$, CI = 19 to 59%; *ZrDepression*: $I^2_{\mathrm{Total}} = 97\%$, CI = 94 to 99%, electronic supplementary material, table S2).

### (b) The strength of inbreeding avoidance depends on the risk of inbreeding depression

In most species where inbreeding depression has been reported, mated pairs were less related than the population

average (22 species out of 34 are below the 1 : 1 line in figure 1b and electronic supplementary material, figure S6). In fact, across all species where inbreeding depression has been found, relatedness between mates was low (figure 2; electronic supplementary material, table S4; Model 3: BPMM: PM = 0.21, CI = −0.22 to 0.69, pMCMC = 0.18, tested against 1: difference = −0.79, CI = −1.22 to −0.31, pMCMC = 0.001). Even in species where the average relatedness in the population was high, such as cooperatively breeding species, we found no evidence for increased levels of inbreeding (electronic supplementary material, table S9; Model 8: BPMM: Difference between cooperative and non-cooperative species = −0.02, CI = −1.41 to 1.35, pMCMC = 0.49, cooperative breeders difference from 1 = −0.91, CI = −2.24 to 0.42, pMCMC = 0.081; non-cooperative species difference from 1 = −0.88, CI = −1.42 to −0.3, pMCMC = 0.003).

In contrast, in species where inbreeding depression has not been reported, mates were as related, or even more closely related, than to the population average (five species out of seven are above the 1 : 1 in figure 1b): Relatedness between pairs was positively correlated with average relatedness in the population and did not statistically significantly differ from one (figures 1b and 2; electronic supplementary material, table S4. BPMM Model 3: PM = 0.55, CI = −0.88 to 2.03, pMCMC = 0.25, difference of slope from 1; PM = −0.45, CI = −1.88 to 1.03, pMCMC = 0.29). This indicates that for species where there is no evidence of inbreeding depression, individuals do not avoid mating with relatives. However, it is important to note that there are a limited number of species for which there is no evidence of inbreeding depression (n = 7). Species with and without evidence of inbreeding depression were widely distributed across the phylogeny showing inbreeding avoidance and inbreeding depression are relatively labile over evolutionary time (figure 1). It is also possible that inbreeding depression has been reduced in some species as a result of purging deleterious alleles [28,52].

## (c) Mechanisms of inbreeding avoidance only evolve when there is a risk of inbreeding depression

In species reported to suffer from inbreeding depression ($N_{Species}$ = 34), avoidance of inbreeding through mate choice (26% of species) was as common as sex-biased dispersal (26% of species), and both mechanisms were more common than post-mating avoidance (15% of species). These mechanisms of inbreeding avoidance were only observed in species reported to suffer from inbreeding depression (figure 2; electronic supplementary material, table S5; Model 4: BPMM: difference between cooperative and non-cooperative species = −0.91, CI = −1 to 0, pMCMC = 0.002). In contrast, mechanisms of inbreeding avoidance were never reported in species without inbreeding depression ($N_{Species}$ = 7). One study on the ground tit *Parus humilis* observed active mate choice based on relatedness for a preferentially mating with relatives [12]. The evolution of all types of inbreeding avoidance mechanisms therefore appears to rely exclusively on the presence of inbreeding depression, although the restricted sample sizes highlight the need further data collection particularly in species without inbreeding depression.

## (d) Random mating does not necessarily increase the risk of inbreeding

Different mechanisms of inbreeding avoidance were equally effective at reducing relatedness between pairs among species with inbreeding depression. Species where mate choice was random with respect to relatedness, were equally successful at avoiding inbreeding as species where mating with relatives was actively avoided (figure 2; electronic supplementary material, figure S8, table S6; Model 5; BPMM: differences between active mate choice and sex-biased dispersal PM = −0.28, CI = −1.9 to 1.33, pMCMC = 0.353; active mate choice and post-mating avoidance PM = 0.12, CI = −1.59 to 1.49, pMCMC = 0.43; sex-biased dispersal and post-mating avoidance PM = −0.4, CI = −1.82 to 1.11, pMCMC = 0.29). Contrary to expectation, we did not find that species with sex-biased dispersal had a lower risk of encountering relatives (electronic supplementary material, table S7; Model 6, figure S9): species with different mechanisms of inbreeding avoidance did not significantly differ in their average relatedness among potential mates (BPMM: difference between sex-biased dispersal and active mate choice PM = 0.11, CI = −0.04 to 0.24, pMCMC = 0.068, difference between sex-biased dispersal and post-mating avoidance PM = −0.04, CI = −0.19 to 0.11, pMCMC = 0.26, difference between sex-biased dispersal and no avoidance PM = 0.08, CI = −0.05 to 0.2, pMCMC = 0.26). However, we did find that species with sex-bias dispersal and post-mating avoidance were more likely to choose their mates at random with respect to relatedness (figure 3; electronic supplementary material, table S8; Model 7; sex-biased dispersal PM = 0.58, CI = 0 to 1, pMCMC = 0.42, post-mating avoidance PM = 0.34, CI = 0 to 1, pMCMC = 0.42, active mate choice PM = 0.01, CI = 0 to 0, pMCMC = 0.01, no avoidance PM = 0.89, CI = 0 to 1, pMCMC = 0.11).

In some species with inbreeding depression, mechanisms of inbreeding avoidance were not reported ($N_{Species}$ = 10), a phenomenon often referred to as an 'inbreeding paradox' [53,54]. Relatedness between partners in these species, although higher, was not statistically significantly different from species where mechanisms of inbreeding avoidance were known (electronic supplementary material, figure S8, table S6, Model 5, mean relatedness between breeding pairs ± s.e.: none = 0.033 ± 0.014, active mate choice = −0.0069 ± 0.014, post-mating avoidance = 0.0082 ± 0.009, sex-biased dispersal = 0.052 ± 0.024). Although data are limited and results not statistically significant, species without known mechanisms of inbreeding avoidance were also at a lower risk of encountering related mates in the population (electronic supplementary material, figure S9, table S7, Model 6, mean relatedness between all pairs in the population ± s.e.: none = 0.019 ± 0.016, active mate choice = 0.058 ± 0.041, post-mating avoidance = 0.061 ± 0.044, sex-biased dispersal = 0.107 ± 0.034). Such trends indicate the need for more data on species with varying degrees of inbreeding depression, but suggest that species which are less likely to meet related mates lack ways of avoiding inbreeding, increasing the chances of inbreeding if relatives are encountered.

## 4. Discussion

Patterns of inbreeding avoidance across species show there is little justification for the notion that a lack of inbreeding avoidance in natural populations is surprising, or counter to

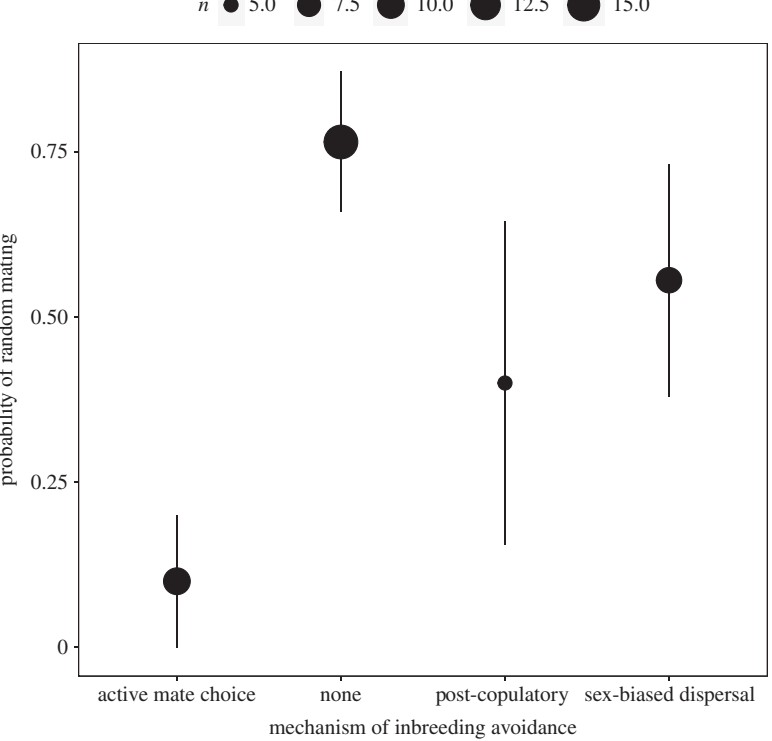

**Figure 3.** Probability of random mating with respect to relatedness in relation to different mechanisms of inbreeding avoidance (active mate choice ($N_{Species} = 10$), none ($N_{Species} = 16$), post-mating avoidance ($N_{Species} = 5$) and sex-biased dispersal ($N_{Species} = 9$). Points represent means with 95% confidence intervals and the size of the point represents $N_{Species}$.

predictions based on evolutionary theory [2,4,5,8]. Animals choose mates based on relatedness when two conditions are fulfilled: when there is a risk related mates encounter each other and when there is inbreeding depression (figure 2). If these conditions are not met, selection for mate choice based on kinship will be weak, even when inbreeding depression is extremely costly. In species without inbreeding depression, the relatedness between pairs tended to increase with relatedness to potential mates, suggesting inbreeding avoidance does not occur (figure 2). In some species, there may even be a net benefit of mating with relatives [13,25,26].

Active mate choice did not offer any measurable advantage over other methods of avoiding inbreeding. Individuals can avoid mating with relatives by actively discriminating on the basis of relatedness (either by mate choice or post-mating avoidance) or by reducing the risk of encountering related mates in the population [17]. In species where the likelihood of encountering related mates in the population was reduced (e.g. dispersal from the natal territory), mate choice was more likely to be random. The apparent lack of avoiding mating with relatives in some species may therefore be explained by there being a negligible risk of inbreeding. Alternatively, in some species, past inbreeding may have resulted in a selective decrease in rare, deleterious and recessive alleles that cause inbreeding depression [28,52], a phenomenon known as purging [52], as has been shown in Chillingham cattle (*Bos taurus*) [55]. Thus, potential purging could also explain a lack of inbreeding depression and avoidance in some species.

Multiple mechanisms of inbreeding avoidance were not reported in the same species, suggesting that each mechanism might be effective on its own. It should be noted, however, that this could be a by-product of the fact that few studies

investigate multiple mechanisms of inbreeding avoidance. One exception is a study by Aguilera-Olivares *et al.* [56], which investigated two different mechanisms of inbreeding avoidance in drywood termites (*Neotermes chilensis*) and found that sex-biased dispersal is the main mechanism of avoiding mating with kin and that there is little evidence of active mate avoidance. Such studies increase the chance of determining which mechanisms of inbreeding avoidance are more likely to evolve in particular species and why. This may help resolve our unexpected finding that sex-biased dispersal was not associated with an overall reduction in average relatedness. For example, other mechanisms may be in place when sex-biased dispersal fails to reduce the chances of inbreeding. In the cooperatively breeding noisy miner (*Manorina melanocephala*), where there is female-biased dispersal, average relatedness in the population is high, yet relatedness between breeding pairs is low. The authors suggest that the noisy miners may be capable of detecting kin and actively avoid choosing to mate with relatives [57] given that noisy miners have been shown to recognize kin based on vocalizations in different contexts [58].

Selection for adopting a strategy based on reducing encounter rates between relatives is likely to depend on constraints on dispersal. For example, although sex-biased dispersal can reduce the chance of inbreeding [17,18], there are also various costs associated with dispersal, such as energetic costs, increased exposure to predators and risk of failure in finding a suitable breeding site [17,59,60]. In addition, habitat fragmentation, as a result of anthropogenic and climate-induced factors, may limit the ability of species to effectively disperse, as is the case for male koalas [61]. Constraints on dispersing from the natal group are therefore likely to generate selection for avoiding mating with

relatives by active discrimination. This may be the case in cooperative breeders such as meerkats (*Suricata suricatta*), which live in extended family groups. Subordinate meerkats are often directly related to other adults in the group and must either 'sit and wait' for a potential mating partner to turn up or face increased risks of mortality from leaving the group [62,63].

Lack of inbreeding avoidance through mate choice in the animal kingdom is often presented as surprising, even a challenge to our understanding of how species should respond to the costs of inbreeding depression [2,4,5]. More broadly our results show that the strength of selection against deleterious actions is not simply a product of the cost incurred, but the frequency with which it happens. Evidence for the importance of risk on the strength of selection for optimizing behaviour also comes from: (i) kin discrimination, where helping kin has a fitness advantage but actively recognizing kin only occurs when related and unrelated individuals are frequently encountered [64]; (ii) paternal care, where caring for unrelated offspring is costly, but adjustment of care by males is not always needed because cuckoldry is rare [65]; and (iii) sex ratio adjustment, where producing more female offspring is beneficial for single foundresses, but this does not occur in species where multiple females typically lay together [66]. Similarly,

reproducing with relatives can result in severe fitness costs, but active mate choice is only selected when related males and females interact [16].

Data accessibility. The data and code in this study are available from the Dryad Digital Repository: https://doi.org/10.5061/dryad.rfj6q57b0 [67].
The data are provided in electronic supplementary material [68].

Authors' contributions. V.L.P.: conceptualization, data curation, formal analysis, funding acquisition, investigation, methodology, project administration, software, validation, visualization, writing-original draft, writing-review and editing; C.K.C.: data curation, formal analysis, funding acquisition, methodology, software, supervision, validation, visualization, writing-review and editing; A.S.G.: conceptualization, investigation, supervision, validation, writing-original draft, writing-review and editing.

All authors gave final approval for publication and agreed to be held accountable for the work performed therein.

Competing interests. We declare we have no competing interests.

Funding. V.L.P. was supported by DPhil funding from the Biotechnology and Biological Sciences Research Council (grant no. BB/M011224/1). C.K.C. was supported by the Swedish Research Council (grant no. 2017-03880) and the Knut and Alice Wallenberg Foundation (Wallenberg Academy fellowship no. 2018.0138).

Acknowledgements. We thank Jon Slate, Per Smiseth and Alfredo Sánchez-Tójar for their comments on a previous version of this manuscript. We also thank Christopher Woodham for his help with the statistical analysis.

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
