## [Peer Review File · Proceedings of the Royal Society B: Biological Sciences]

Review History

RSPB-2020-2372.R0 (Original submission)

Review form: Reviewer 1 (Alfredo Sánchez-Tójar)

Recommendation

Major revision is needed (please make suggestions in comments)

Scientific importance: Is the manuscript an original and important contribution to its field?

Excellent

General interest: Is the paper of sufficient general interest?

Good

Quality of the paper: Is the overall quality of the paper suitable?

Marginal

Is the length of the paper justified?

Yes

Should the paper be seen by a specialist statistical reviewer?

No

Do you have any concerns about statistical analyses in this paper? If so, please specify them explicitly in your report.

Yes

It is a condition of publication that authors make their supporting data, code and materials available - either as supplementary material or hosted in an external repository. Please rate, if applicable, the supporting data on the following criteria.

Is it accessible?

Yes

Is it clear?

Yes

Is it adequate?

Yes

Do you have any ethical concerns with this paper?

No

Comments to the Author

In this study, the authors used a meta-analytic approach to test multiple hypotheses about inbreeding avoidance, including under which circumstances inbreeding avoidance is expected to preferentially evolve. To do so, they compiled a dataset containing 48 effect sizes (47 animal species). The manuscript is well written and the hypotheses are very interesting, however I have some major methodological concerns about the literature search, the data extraction and the statistical analyses. My main concerns and suggestions are described below followed by additional line-to-line comments. I apologize for the length of my review, but I hope my comments will be helpful in revising this contribution.

Literature search: my main concerns about the literature search are: (1) the lack of reporting essential information to understand what was done and to allow reproducibility; and (2) the potential incompleteness and, hopefully to a lesser extent, bias of the search strategy used.

(1) It is unclear what specific search string was used and which databases within Web of Science (WoS) were searched. Presumably the authors searched WoS Core Collection and did a topic (TS) search (title, abstract, keywords), which depending on the authors' subscription would cover more or less databases (see "Citation Indexes" in tab "Advanced Search"). Regarding the search string, I would assume the authors searched for "(inbreeding avoidance OR incest avoidance)" rather than "(inbreeding avoidance AND incest avoidance)". However, the reported "N ~ 1,400" in Figure S1 would indicate that the authors indeed searched for studies containing both keywords "inbreeding avoidance" AND "incest avoidance" - my own searching of studies containing "inbreeding avoidance" OR "incest avoidance" led to 1381 hits from WoS Core Collection and 13,200 hits from Google Scholar. Additionally, Figure S1 would require some corrections. I don't think it is good practice to provide approximate values because the purpose of the PRISMA diagram is to increase the reproducibility of the literature search, and it's unclear how approximated values would help with that. Therefore, I would lean towards stating that those values are unknown, if that is the case. Furthermore, numbers do not seem to add up for the fulltext screening and included steps (e.g. number of papers screened and excluded is the same). Last, did the authors screen ~27000 titles and abstracts manually or used automatic procedures such as machine learning? Overall, although not always reported in evidence synthesis in ecology and evolution, all these details are important to increase reproducibility in evidence synthesis (e.g. Sánchez-Tójar et al. 2020b; Haddaway et al. 2020), so I would suggest to provide as many as possible and acknowledge those that cannot be provided as limitations.

(2) Assuming the authors searched for "(inbreeding avoidance OR incest avoidance)", this strategy would seemingly miss presumably important alternative keywords such as "(avoid* inbreeding OR avoid* incest)" (and potentially more combinations). Indeed, simply adding

“(“avoid* inbreeding” OR “avoid* incest”)” to the search string leads to 15% increase in hits in WoS Core Collection (200+ references) and potentially hundreds more hits from Google Scholar. In addition, using “(“inbreeding avoidance” AND “incest avoidance”)” could be biased towards finding only papers that found evidence for avoidance but not for preference (or no association). My suggestion would therefore be to also include “preference” in the search string, e.g. “(“inbreeding preference” OR “incest preference”)”, which will find a few extra hits that could be important (e.g. Thunken et al. 2011, Lange et al. 2017). The two potential problems highlighted in point (2) might have been mitigated thanks to the forward and backward searches performed by the authors, however, whether that was the case would require confirmation in order to evaluate the quality of the search strategy.

Data extraction: my main concerns about data extraction are whether: (1) average relatedness between potential mates is comparable across studies; and (2) the effect size chosen in the correct one. I do not have a definite answer about these two points, but would like to hear the authors’ thoughts. Also, I’m not an expert on inbreeding avoidance, so I apologize in advance for any misunderstanding.

(1) I don’t understand very well what the “average relatedness between potential mates in the population” represents across studies. I tried to understand it better by looking at three included studies but there seem to be some conceptual differences in how that average relatedness was calculated across them, and I wonder whether those differences should be taken into account in the analyses. For example, the values extracted from Barati et al. (2018) seem to correspond to genetic relatedness for breeding pairs (ca. 0.05) and for breeding females and their helpers (ca. 0.18; Figure 1), whereas the values extracted from Cayuela et al. (2017) seem to correspond to genetic relatedness for males and females from the same (0.21) and different breeding patches (0.04); and the values extracted from Griffin et al. (2003) seem to correspond to genetic relatedness for breeding pairs (ca. 0.05) and all pairs in the population (0.23; Figure 5). A priori, I was expecting that the average relatedness between potential mates in the population would be always calculated as in the latter example (Griffin et al. 2003), and I wonder whether the other strategies are indeed comparable. Perhaps the author could explain this further.

(2) Can estimates of relatedness really be considered correlation coefficients rather than mean proportions/percentages of similarity across pairs? I have doubts whether it is conceptually valid to treat those means as correlation coefficients, so perhaps the authors could explain this further. I wonder whether for the main purposes of this meta-analysis, calculating a mean ratio/difference (e.g. lnRR or SMDH) between mean relatedness between pairs and average relatedness would be a more appropriate effect size. This would also allow the authors to accommodate the differences in uncertainty (e.g. SE) between estimates of ZrParis and ZrAverage, and to take into account the sampling variance of ZrAverage, which so far it is only approximately taken into account by using log(n)? (see comment about log(n) below). If the authors decide to treat them as correlations, they should be aware that, due to the [0,1] bounding, Fisher’s z-transformations will not help much with normalizing the distribution of the data, and so, it might be a good idea to run sensitivity analyses using beta regressions (as done in Doehrmann et al. 2019).

Statistical analyses: my main concerns about the statistical analyses are: (1) the calculation of I2 for meta-regressions instead of R2marginal; and (2) the validity of the publication bias tests performed.

(1) It is not very intuitive to calculate – or better said, to call it, “heterogeneity I2” in meta-regressions. I2 is a relative measure of heterogeneity that is normally calculated for meta-analytic (i.e. intercept-only) models to show whether heterogeneity is present (and how much and from where it comes from), and thus, to inform us whether meta-regressions should be implemented to try and explain heterogeneity, if any found (Senior et al. 2016). Therefore, it is uncommon to estimate I2 from meta-regressions, but R2marginal instead (Nakagawa and Schielzeth 2013), because the latter gives us an approximation of the percentage of heterogeneity observed in the meta-analytic (intercept-only) model explained by the moderator(s) included in the subsequent meta-regression. Estimating I2 from a meta-regression would be equivalent to estimating adjusted repeatability (Nakagawa and Schielzeth 2010), and as such, we should consider whether the variance explained by the moderators (R2marginal) needs to be added to the denominator

when estimating I_2 , which depends on what we want I_2 to reflect (Nakagawa and Schielzeth 2010). Overall, my preference would be to calculate and discuss R^2_{marginal} for all meta-regressions (see code from Sánchez-Tójar et al. 2018), and to simply report the non-standardized variance components from the meta-regressions; although the authors could consider providing H^2 (i.e. I_2 phylogeny; sensu Nakagawa and Santos 2012) and discuss the biological importance of those results.

(2) Regarding the publication biases tests, I don't understand well what was done, particularly since the code for these analyses was seemingly not provided. From the description and the plots presented (Figures S8-S10), I suspect the publication bias analyses might not have been conducted appropriately. As far as I know, `regtest()` and `trimfill()` are not yet implemented for multilevel models. Also, I don't understand why the x-axis of Figures S8-S10 says log odds ratio and not Z_r (i.e. Fisher's Z), and I don't seem to find the trim and fill numeric results anywhere. To properly test for publication bias in multilevel models, the random effects variance should to be taken into account either by extracting meta-analytic residuals (more in Nakagawa and Santos 2012; see code from Sánchez-Tójar et al. 2018 – depending on the version of MCMCglmm used) or perhaps easier making use of multilevel meta-regressions that include precision (i.e. the inverse of the sampling variance) as a moderator. That would correspond with an adjusted Egger's test that would show whether effect sizes approximate zero with an increase in precision (see example in section 5.6 from supplementary material in Sánchez-Tójar et al. 2020b). I would also recommend the authors to test for time-lag bias by including "year of publication" as moderator (e.g. Sánchez-Tójar et al. 2018; reviewed in Koricheva and Kulinskaya 2019). Last, as for the trim and fill method, I would actually suggest not running it as this method does not seem to work very well when heterogeneity is present, and overall its results can be misleading since it is not possible to know if and how many effect sizes remained unpublished.

PS: Please, consider citing all the references of the included studies in the main text so that primary researchers get the credit they deserve. If the limitation is how many references Proceedings B allows to be cited, I would like to take advantage and ask Proceedings B to allow the authors to cite all references included in their analyses. Other journals allow so (e.g. Biological Reviews, Nature Ecology and Evolution) and there is no clear reason why Proceedings B should not adopt this when publishing evidence synthesis.

Line-to-line comments:

Line 60-61: I wonder whether MHC-dependent mate selection would be a specific type of inbreeding avoidance/preference worth integrating (meta-analysis by Winternitz et al. 2016).

Line 74: probably also highly variable across populations of the same species?

Line 84: I would use "phylogenetic meta-analysis" here and throughout (e.g. the abstract) to use consistent terminology and make it easier for the reader.

Line 93-94: something that was not explicitly estimated was that variation in the strength. I think that before running the meta-regressions and testing all these hypotheses, the authors could run a Bayesian phylogenetic multilevel meta-analysis (i.e. intercept-only model) to show what is the overall effect size for Z_r Pairs (although see effect size suggestion), it's 95% Credible Intervals, it's 95% Prediction intervals (more in Nakagawa et al. 2020; Sánchez-Tójar et al. 2020b), and the absolute (Q test) and relative (I_2) heterogeneity. Then, the meta-regressions would aim at explaining the heterogeneity found in the intercept-only model. Just a suggestion that depends also on the final effect size of choice.

Line 104-105: if possible, please, provide the full list of studies found by the searches to increase reproducibility.

Line 105-106: does "all" refer to the 287 studies included in fulltext screening? Could you please provide in the supplements the list of studies for which forward and backward searches were performed? Was this process done manually or automatically? Also, do I understand correctly that only studies on wild populations were included in this meta-analysis? If so, I would expect to see that criterion appear in the list of exclusion reasons provided in Table S5.

Line 113-134: since different approaches to estimate relatedness (pedigree vs. markers vs. genome-wide) can lead to more or less precise estimates, I would consider testing if estimates differ depending on the method used (assuming there were more than one method use across

studies). Additionally, estimates of relatedness can also be more or less precise depending on the quality of, for example, the pedigree, so I would consider exploring whether those potential methodological differences might explain some of the variance in results across studies. That is, effect sizes here likely vary depending on both sampling variance (i.e. number of individuals) and genetic information to reconstruct relatedness.

Line 122-124: as far as I can see in the data, sample sizes of both populations are the same (N=270; is that correct?) meaning that a weighted and non-weighted mean would be in principle the same?

Line 141-142: did the authors perform additional searches to find more information about these mechanisms from other studies, just to make sure there is agreement across the literature for each species? Does this include only papers included in the final analysis (n = 48) or all the others (n=287; Table S5)? I guess there could be important information for these classifications in all papers listed in Table S5.

Line 148: doesn't this assume that extra-pair copulations' are a strategy of inbreeding avoidance even when this might not be necessarily true across species?

Like 149-150: this doesn't sound optimal; shouldn't they instead be classified as "NA" or perhaps "unclear" to still be able to use the data?

Line 154: when discussing these results, it might be interesting to compare them with that of previous meta-analyses on this topic: Crnokrak and Roff 1999 (wild organisms), Leroy 2014 (livestock), and Clark et al. 2019 (humans)

Line 155-158: please, report which statistics were encountered (F-tests, Spearman rank correlations...) and which equations (or where they came from) were used to calculate r? What did the authors do if they found more than one type of statistic for the same analysis? Did the authors use an a priority list to choose among them? Last, did the authors extract data from plots (e.g. using the R package metaDigitise; Pick et al. 2019), it seems so, but I did not find this information in the methods.

Line 169-172: how more likely? I would expect this topic to particularly suffer from publication bias, particularly depending on the method used to estimate inbreeding (see comment above). Did the authors test so? (see recommendations below).

Line 185-186: I recommend calling the analyses "Bayesian phylogenetic multilevel meta-regressions" to make it clear when effect sizes were weighted by the inverse of the sampling variance. I personally had to look at the code to understand this.

Line 187: please, provide the version of all R packages for reproducibility purposes (Pasquier et al. 2017).

Line 191-192: "1/(n-3)" instead?

Line 194-196: as far as I can see from the code, rather than the number of individuals in the population, what is used is the number of paired individuals. Wouldn't this number generally be different to the number of individuals used to estimate $Z_{rAverage}$?

Line 205-206: shouldn't the wording rather be something like: "if evidence for inbreeding avoidance differ between species with and without inbreeding depression"? Also, "statistically significantly" (and throughout).

Line 215: Figure 3 indicates 39 data points rather than 38. Is that correct?

Line 243-244: from the code it seems that the authors randomly chose one model of the three, so I would state that here.

Line 298-299: I'm really not sure how appropriate this number is since it is a mean of means, and therefore uncertainty of the original means is neglected. It might be better to provide meta-analytic means here.

Line 311: what is "PM"? Apologies if I missed it.

Line 336-339: but the effect is not that far from statistical significance, so I would interpret these results with caution (as done for some other results shown below; more on this right below).

Line 373-376: I personally prefer this interpretation (i.e. prefer to avoid hard dichotomies), but the authors would need to adjust their interpretations of small but not statistically significant p-values throughout to be consistent - in this example CI's overlap with 0 and p-value is not < 0.05.

Line 380: please report the estimates or refer to the table that contains them.

Line 393-395: I would suggest to interpret these results more cautiously since only 7 data points were available for that specific regression, and despite non-statistical significance, the regression

tends to be less steep than 1:1

Line 395-396: see my comments about considering this scenario when searching for studies (i.e. adding “preference” to the search string). Is Figure S2 correct here?

Line 408: I would suggest “might not be effective on its own”. As suggested in the next sentence, I don’t think we can be sure that if a mechanism is not reported for a species, that means it does not happen.

Line 429: the following reference seems relevant here (and throughout): Avilés and Purcell 2012

Line 451: I miss a paragraph discussing the potential limitations of the study (e.g. relatively few effect sizes, etc), what heterogeneity means for the interpretation, etc.

Line 461: “The data and code in this study”

Figure 1: I would recommend to add uncertainty for both axis to each dot (e.g. Figure 3 from Winney et al. 2018) so that the interpretation (e.g. line 325 and 335-336) is easier and more precise.

Figure 3: is the y-axis wording correct?

Table 2: consider showing residual variance for non meta-analytic models too.

Supplementary material

Minor: Olson et al. 2012 reference misses the last author, I believe.

Table S5: there seem to be duplicated rows.

Data

Please add the metadata provided in the supplementary material to the code and/or to an extra tab (or readme) in the excel file. Furthermore, please provide more bibliographic information (e.g. doi) in the dataset for the references included (and the excluded ones from Table S5) so that a reader can easily find them without having to match the supplementary material pdf with the dataset. All that would increase data reusability.

Overall, I would recommend a little bit of more data cleaning even if that will not change any results but just to increase standardization as much as possible. For example, remove unnecessary spaces in some levels (e.g. “cincta”, “reticulata”, see also variables “inbreeding_avoidance” and “Reported_in_study”), use the same format throughout (e.g. references, capital first letter for common_name), etc.

Cayuela et al., 2017: shouldn’t the sample size be 50 rather than 60? Did the authors perform a data extraction double-check? If not, I would recommend to perform a double-check of a percentage of the data to confirm that data is generally correct (e.g. 25% as in Moran et al. 2020).

Code

The format of the dataset provided is .xlsx, whereas the code tries to import a .csv file. Please, confirm that the data provided is the correct one. I assumed so for my review, however, I got the following error for lines 47-49: “Error in `[.data.frame](data, rh.idx, ids.var, drop = FALSE) : undefined columns selected”, which seems to contradict my assumption since the column “obs_or_expt” is missing.

As a general comment, I would recommend adding more inline comments to the code to make it easier to understand it. For example, I cannot easily follow the “weighted average” performed from line 41 on.

Code line 86-88: I believe this check shows that the ott_id for “simochromis pleurospilus” should be 474353 instead of 710012?

Code line 301: “Method_of_IA2” does not seem to be defined beforehand in the code, and as such, “graph_1.b” cannot be recreated. Please, revise accordingly.

Model M5 does not seem to converge nor run well, particularly the random effect part of the model, but also a little bit the fixed effect part. Plus, the effective sampling is very low, actually 0 for “units”. Did the authors had similar issues?

Overall, I would recommend the authors to save and provide the models they ran (e.g. save them as .rda) to increase reproducibility, and also to make it easier for researchers that want to run the code without having to wait for all the models.

Last, I feel a bit odd saying this and I might as well be totally wrong, if so, I wholeheartedly apologize, but I seem to recognize chunks of code as code I wrote for Sánchez-Tójar et al. 2020a (code: https://github.com/ASanchez-Tojar/meta-analysis_of_variance) and seemingly also

Sánchez-Tójar et al. 2018 (code: https://github.com/ASanchez-Tojar/meta-analysis_sparrows_ssh)? If so, and since incentives are important in academia, I would appreciate if the authors would cite the original studies (Sánchez-Tójar et al. 2018, 2020a), the same way as we would normally do when reusing data or ideas from previous publications. If not, again, I apologize.

Sincerely, Alfredo Sánchez-Tójar. Note that I sign all my reviews since March 2018. I am more than happy to discuss any of the suggestions I have made with the authors and/or editors (my email is alfredo.tojar@gmail.com)

References:

- Avilés and Purcell 2012: <https://doi.org/10.1016/B978-0-12-394288-3.00003-4>
 Clark et al. 2019: <https://www.nature.com/articles/s41467-019-12283-6>
 Crnokrak and Roff 1999: <https://www.nature.com/articles/6885530>
 Dochtermann et al. 2019: <https://doi.org/10.1093/jhered/esz023>
 Duthie and Reid 2016: <https://doi.org/10.1086/688919>
 Haddaway et al. 2020: <https://t.co/QRGy3jjAuc?amp=1>
 Koricheva and Kulinskaya 2019: <https://doi.org/10.1016/j.tree.2019.05.006>
 Lange et al. 2017: <https://doi.org/10.1093/biolinnean/blw003>
 Leedale et al. 2020: <https://doi.org/10.1073/pnas.1918726117>
 Leroy 2014: <https://doi.org/10.1111/age.12178>
 Moran et al. 2020: <https://doi.org/10.1111/brv.12655>
 Nakagawa and Schielzeth 2010: <https://doi.org/10.1111/j.1469-185X.2010.00141.x>
 Nakagawa and Schielzeth 2013: <https://doi.org/10.1111/j.2041-210x.2012.00261.x>
 Nakagawa et al. 2020: <https://doi.org/10.1002/jrsm.1424>
 Pasquier et al. 2017: <https://doi.org/10.1038/sdata.2017.114>
 Pick et al. 2019: <https://doi.org/10.1111/2041-210X.13118>
 Sánchez-Tójar et al. 2018: <https://doi.org/10.7554/eLife.37385>
 Sánchez-Tójar et al. 2020a: <https://doi.org/10.1111/jeb.13661>
 Sánchez-Tójar et al. 2020b: <https://doi.org/10.1111/ele.13479>
 Senior et al. 2016: <https://doi.org/10.1002/ecy.1591>
 Thunken et al. 2011: <https://doi.org/10.1093/beheco/arq217>
 Wells et al. 2020: <https://doi.org/10.1111/ele.13578>
 Winney et al. 2018: <https://doi.org/10.1111/jeb.13197>
 Winternitz et al. 2016: <https://doi.org/10.1111/mec.13920>

Review form: Reviewer 2 (Per Smiseth)

Recommendation

Major revision is needed (please make suggestions in comments)

Scientific importance: Is the manuscript an original and important contribution to its field?

Good

General interest: Is the paper of sufficient general interest?

Excellent

Quality of the paper: Is the overall quality of the paper suitable?

Good

Is the length of the paper justified?

No

Should the paper be seen by a specialist statistical reviewer?

No

Do you have any concerns about statistical analyses in this paper? If so, please specify them explicitly in your report.

No

It is a condition of publication that authors make their supporting data, code and materials available - either as supplementary material or hosted in an external repository. Please rate, if applicable, the supporting data on the following criteria.

Is it accessible?

N/A

Is it clear?

N/A

Is it adequate?

N/A

Do you have any ethical concerns with this paper?

No

Comments to the Author

This paper addresses a topic of great general interest to biologists; that is, the evolution of inbreeding avoidance mechanisms such as mating preferences for unrelated partners. My main concern is that the paper presents a simple linear causal relationship between inbreeding depression and inbreeding avoidance, where the former selects for the latter. I completely agree that the severity of inbreeding depression is likely to be a strong selective force on inbreeding avoidance. However, the causal relationship between the two is likely to be more complex as patterns of inbreeding in the past also will determine the severity of inbreeding depression. For example, in species that have experienced frequent inbreeding in the past, there may have been purging of the rare, recessive and deleterious alleles that cause inbreeding depression. Thus, the association between inbreeding avoidance and inbreeding depression is likely to reflect a more complex causal relationship. I don't think this makes the results any less interesting, but I suggest that the authors revise some of their predictions (see specific comment #5) and some of the interpretation of their results (see specific comment #7).

Specific comments:

- (1) Lines 47–48: Please clarify what you mean by inbreeding? Inbreeding may refer to the process of mating in parental generation and/or it may refer to the production of inbred individuals in the offspring's generation. Selection for inbreeding avoidance would obviously happen at the parents' generation, whilst the fitness consequences are felt at the offspring's generation. There are also several definitions of inbreeding in the literature. For example, it may refer to mating between related individuals or non-random mating where related individuals are more likely to mate than expected by chance. For clarity, it might be useful if you state how you define inbreeding. Just to take an example where these definitions would matter: assume a small population consisting of a single male and a single female who were brother and sister. This is inbreeding according to a pedigree-based definition but not according to a definition based on random mating.
- (2) Line 69: Is kin recognition here the same as inbreeding avoidance by mate choice? Please clarify.
- (3) Lines 155–158: Is this your own metric for severity of inbreeding depression? If so, make this clear to the reader. If not, provide a reference for how it has been used in the past.
- (4) Lines 158–160: Please clarify whether the focal individual that is inbred or outbred corresponds with the individual whose traits you refer to here. For example, it is unclear here

whether offspring mortality and offspring mass refers to the mass of inbred offspring or the offspring of inbred parents?

(5) Lines 201–205: I'm not convinced by this argument. This relationship may also be less than one if the absence of inbreeding avoidance has led to frequent inbreeding in the past and thereby purging of rare, deleterious and recessive alleles. See also major comment.

(6) Lines 299–301: Please check this statement. It is unclear to me how relatedness can be negative if it is bounded between 0 and 1 (see lines 130–133)?

(7) 339–343: An alternative explanation here is that the lack of inbreeding avoidance has been associated inbreeding in the past, which has purged the population of the rare, deleterious and recessive alleles that cause inbreeding depression. See also major comment.

(8) Line 372: Please check this subheading and similar statements throughout. I appreciate that this is not your intention, but this might be read as a 'good for the species' argument.

(9) Lines 380–382: I'm not sure if I get this argument. Even if a given species has evolved inbreeding avoidance mechanisms, it seems unlikely that such mechanisms would be perfect. If not, there should be some risk of inbreeding and if this is the case, should we not expect inbreeding depression to be more severe when parents are more related? Is this finding robust or does it reflect low statistical power due to few instances of inbreeding between closely related parents?

(10) Lines 395–396: Consider rewording this statement. I would argue that there is always some transmission benefit (or inclusive fitness benefit) of inbreeding as it increases the probability that a given gene is transmitted. However, this benefit is often outweighed by the cost due to the reduction in fitness of inbred offspring. Presumably, this statement refers to the net benefit of inbreeding?

(11) Line 399: Individuals avoid inbreeding (not species).

(12) Lines 439–440: Please add a reference to support this claim.

Per Smiseth (I always sign my reviews)

Decision letter (RSPB-2020-2372.R0)

06-Nov-2020

Dear Miss Pike:

I am writing to inform you that your manuscript RSPB-2020-2372 entitled "Why don't all animals avoid inbreeding?" has, in its current form, been rejected for publication in Proceedings B.

This action has been taken on the advice of referees, who have recommended that substantial revisions are necessary. With this in mind we would be happy to consider a resubmission, provided the comments of the referees are fully addressed. However please note that this is not a provisional acceptance.

Sincerely,
 Dr Sasha Dall
 mailto: proceedingsb@royalsociety.org

Associate Editor
 Board Member: 1
 Comments to Author:

The authors performed a meta analysis and investigated inbreeding avoidance across the animal kingdom. The topic is very interesting and I enjoyed reading the manuscript.

At the same time I feel that the methodology part is missing some cubical information, i.e. how was the literature used in the meta analysis found, i.e. exact search string as pointed out by one of the referees.

Reviewer(s)' Comments to Author:

Referee: 1

Comments to the Author(s)

In this study, the authors used a meta-analytic approach to test multiple hypotheses about inbreeding avoidance, including under which circumstances inbreeding avoidance is expected to preferentially evolve. To do so, they compiled a dataset containing 48 effect sizes (47 animal species). The manuscript is well written and the hypotheses are very interesting, however I have some major methodological concerns about the literature search, the data extraction and the statistical analyses. My main concerns and suggestions are described below followed by additional line-to-line comments. I apologize for the length of my review, but I hope my comments will be helpful in revising this contribution.

Literature search: my main concerns about the literature search are: (1) the lack of reporting essential information to understand what was done and to allow reproducibility; and (2) the potential incompleteness and, hopefully to a lesser extent, bias of the search strategy used.

(1) It is unclear what specific search string was used and which databases within Web of Science (WoS) were searched. Presumably the authors searched WoS Core Collection and did a topic (TS) search (title, abstract, keywords), which depending on the authors' subscription would cover more or less databases (see "Citation Indexes" in tab "Advanced Search"). Regarding the search string, I would assume the authors searched for "(inbreeding avoidance" OR "incest avoidance")" rather than "(inbreeding avoidance" AND "incest avoidance)". However, the reported "N ~ 1,400" in Figure S1 would indicate that the authors indeed searched for studies containing both keywords "inbreeding avoidance" AND "incest avoidance" - my own searching of studies containing "inbreeding avoidance" OR "incest avoidance" led to 1381 hits from WoS Core Collection and 13,200 hits from Google Scholar. Additionally, Figure S1 would require some corrections. I don't think it is good practice to provide approximate values because the purpose of the PRISMA diagram is to increase the reproducibility of the literature search, and it's unclear how approximated values would help with that. Therefore, I would lean towards stating that those values are unknown, if that is the case. Furthermore, numbers do not seem to add up for

the fulltext screening and included steps (e.g. number of papers screened and excluded is the same). Last, did the authors screened ~27000 titles and abstracts manually or used automatic procedures such as machine learning? Overall, although not always reported in evidence synthesis in ecology and evolution, all these details are important to increase reproducibility in evidence synthesis (e.g. Sánchez-Tójar et al. 2020b; Haddaway et al. 2020), so I would suggest to provide as many as possible and acknowledge those that cannot be provided as limitations.

(2) Assuming the authors searched for “(“inbreeding avoidance” OR “incest avoidance”)”, this strategy would seemingly miss presumably important alternative keywords such as “(“avoid* inbreeding” OR “avoid* incest”)” (and potentially more combinations). Indeed, simply adding “(“avoid* inbreeding” OR “avoid* incest”)” to the search string leads to 15% increase in hits in WoS Core Collection (200+ references) and potentially hundreds more hits from Google Scholar. In addition, using “(“inbreeding avoidance” AND “incest avoidance”)” could be biased towards finding only papers that found evidence for avoidance but not for preference (or no association). My suggestion would therefore be to also include “preference” in the search string, e.g. “(“inbreeding preference” OR “incest preference”)”, which will find a few extra hits that could be important (e.g. Thunken et al. 2011, Lange et al. 2017). The two potential problems highlighted in point (2) might have been mitigated thanks to the forward and backward searches performed by the authors, however, whether that was the case would require confirmation in order to evaluate the quality of the search strategy.

Data extraction: my main concerns about data extraction are whether: (1) average relatedness between potential mates is comparable across studies; and (2) the effect size chosen in the correct one. I do not have a definite answer about these two points, but would like to hear the authors’ thoughts. Also, I’m not an expert on inbreeding avoidance, so I apologize in advance for any misunderstanding.

(1) I don’t understand very well what the “average relatedness between potential mates in the population” represents across studies. I tried to understand it better by looking at three included studies but there seem to be some conceptual differences in how that average relatedness was calculated across them, and I wonder whether those differences should be taken into account in the analyses. For example, the values extracted from Barati et al. (2018) seem to correspond to genetic relatedness for breeding pairs (ca. 0.05) and for breeding females and their helpers (ca. 0.18; Figure 1), whereas the values extracted from Cayuela et al. (2017) seem to correspond to genetic relatedness for males and females from the same (0.21) and different breeding patches (0.04); and the values extracted from Griffin et al. (2003) seem to correspond to genetic relatedness for breeding pairs (ca. 0.05) and all pairs in the population (0.23; Figure 5). A priori, I was expecting that the average relatedness between potential mates in the population would be always calculated as in the latter example (Griffin et al. 2003), and I wonder whether the other strategies are indeed comparable. Perhaps the author could explain this further.

(2) Can estimates of relatedness really be considered correlation coefficients rather than mean proportions/percentages of similarity across pairs? I have doubts whether it is conceptually valid to treat those means as correlation coefficients, so perhaps the authors could explain this further. I wonder whether for the main purposes of this meta-analysis, calculating a mean ratio/difference (e.g. lnRR or SMDH) between mean relatedness between pairs and average relatedness would be a more appropriate effect size. This would also allow the authors to accommodate the differences in uncertainty (e.g. SE) between estimates of ZrParis and ZrAverage, and to take into account the sampling variance of ZrAverage, which so far it is only approximately taken into account by using log(n)? (see comment about log(n) below). If the authors decide to treat them as correlations, they should be aware that, due to the [0,1] bounding, Fisher’s z-transformations will not help much with normalizing the distribution of the data, and so, it might be a good idea to run sensitivity analyses using beta regressions (as done in Dochtermann et al. 2019).

Statistical analyses: my main concerns about the statistical analyses are: (1) the calculation of I2 for meta-regressions instead of R2marginal; and (2) the validity of the publication bias tests performed.

(1) It is not very intuitive to calculate – or better said, to call it, “heterogeneity I2” in meta-regressions. I2 is a relative measure of heterogeneity that is normally calculated for meta-analytic

(i.e. intercept-only) models to show whether heterogeneity is present (and how much and from where it comes from), and thus, to inform us whether meta-regressions should be implemented to try and explain heterogeneity, if any found (Senior et al. 2016). Therefore, it is uncommon to estimate I^2 from meta-regressions, but R^2_{marginal} instead (Nakagawa and Schielzeth 2013), because the latter gives us an approximation of the percentage of heterogeneity observed in the meta-analytic (intercept-only) model explained by the moderator(s) included in the subsequent meta-regression. Estimating I^2 from a meta-regression would be equivalent to estimating adjusted repeatability (Nakagawa and Schielzeth 2010), and as such, we should consider whether the variance explained by the moderators (R^2_{marginal}) needs to be added to the denominator when estimating I^2 , which depends on what we want I^2 to reflect (Nakagawa and Schielzeth 2010). Overall, my preference would be to calculate and discuss R^2_{marginal} for all meta-regressions (see code from Sánchez-Tójar et al. 2018), and to simply report the non-standardized variance components from the meta-regressions; although the authors could consider providing H^2 (i.e. $I^2_{\text{phylogeny}}$; sensu Nakagawa and Santos 2012) and discuss the biological importance of those results.

(2) Regarding the publication biases tests, I don't understand well what was done, particularly since the code for these analyses was seemingly not provided. From the description and the plots presented (Figures S8-S10), I suspect the publication bias analyses might not have been conducted appropriately. As far as I know, `regtest()` and `trimfill()` are not yet implemented for multilevel models. Also, I don't understand why the x-axis of Figures S8-S10 says log odds ratio and not Z_r (i.e. Fisher's Z), and I don't seem to find the trim and fill numeric results anywhere. To properly test for publication bias in multilevel models, the random effects variance should be taken into account either by extracting meta-analytic residuals (more in Nakagawa and Santos 2012; see code from Sánchez-Tójar et al. 2018 – depending on the version of `MCMCglmm` used) or perhaps easier making use of multilevel meta-regressions that include precision (i.e. the inverse of the sampling variance) as a moderator. That would correspond with an adjusted Egger's test that would show whether effect sizes approximate zero with an increase in precision (see example in section 5.6 from supplementary material in Sánchez-Tójar et al. 2020b). I would also recommend the authors to test for time-lag bias by including "year of publication" as moderator (e.g. Sánchez-Tójar et al. 2018; reviewed in Koricheva and Kulinskaya 2019). Last, as for the trim and fill method, I would actually suggest not running it as this method does not seem to work very well when heterogeneity is present, and overall its results can be misleading since it is not possible to know if and how many effect sizes remained unpublished.

PS: Please, consider citing all the references of the included studies in the main text so that primary researchers get the credit they deserve. If the limitation is how many references Proceedings B allows to be cited, I would like to take advantage and ask Proceedings B to allow the authors to cite all references included in their analyses. Other journals allow so (e.g. Biological Reviews, Nature Ecology and Evolution) and there is no clear reason why Proceedings B should not adopt this when publishing evidence synthesis.

Line-to-line comments:

Line 60-61: I wonder whether MHC-dependent mate selection would be a specific type of inbreeding avoidance/preference worth integrating (meta-analysis by Winternitz et al. 2016).

Line 74: probably also highly variable across populations of the same species?

Line 84: I would use "phylogenetic meta-analysis" here and throughout (e.g. the abstract) to use consistent terminology and make it easier for the reader.

Line 93-94: something that was not explicitly estimated was that variation in the strength. I think that before running the meta-regressions and testing all these hypotheses, the authors could run a Bayesian phylogenetic multilevel meta-analysis (i.e. intercept-only model) to show what is the overall effect size for Z_r Pairs (although see effect size suggestion), it's 95% Credible Intervals, it's 95% Prediction intervals (more in Nakagawa et al. 2020; Sánchez-Tójar et al. 2020b), and the absolute (Q test) and relative (I^2) heterogeneity. Then, the meta-regressions would aim at explaining the heterogeneity found in the intercept-only model. Just a suggestion that depends also on the final effect size of choice.

Line 104-105: if possible, please, provide the full list of studies found by the searches to increase reproducibility.

Line 105-106: does “all” refer to the 287 studies included in fulltext screening? Could you please provide in the supplements the list of studies for which forward and backward searches were performed? Was this process done manually or automatically? Also, do I understand correctly that only studies on wild populations were included in this meta-analysis? If so, I would expect to see that criterion appear in the list of exclusion reasons provided in Table S5.

Line 113-134: since different approaches to estimate relatedness (pedigree vs. markers vs. genome-wide) can lead to more or less precise estimates, I would consider testing if estimates differ depending on the method used (assuming there were more than one method use across studies). Additionally, estimates of relatedness can also be more or less precise depending on the quality of, for example, the pedigree, so I would consider exploring whether those potential methodological differences might explain some of the variance in results across studies. That is, effect sizes here likely vary depending on both sampling variance (i.e. number of individuals) and genetic information to reconstruct relatedness.

Line 122-124: as far as I can see in the data, sample sizes of both populations are the same (N=270; is that correct?) meaning that a weighted and non-weighted mean would be in principle the same?

Line 141-142: did the authors perform additional searches to find more information about these mechanisms from other studies, just to make sure there is agreement across the literature for each species? Does this include only papers included in the final analysis (n = 48) or all the others (n=287; Table S5)? I guess there could be important information for these classifications in all papers listed in Table S5.

Line 148: doesn't this assume that extra-pair copulations' are a strategy of inbreeding avoidance even when this might not be necessarily true across species?

Like 149-150: this doesn't sound optimal; shouldn't they instead be classified as “NA” or perhaps “unclear” to still be able to use the data?

Line 154: when discussing these results, it might be interesting to compare them with that of previous meta-analyses on this topic: Crnokrak and Roff 1999 (wild organisms), Leroy 2014 (livestock), and Clark et al. 2019 (humans)

Line 155-158: please, report which statistics where encountered (F-tests, Spearman rank correlations...) and which equations (or where they came from) were used to calculate r? What did the authors do if they found more than one type of statistic for the same analysis? Did the authors used an a priority list to choose among them? Last, did the authors extract data from plots (e.g. using the R package metaDigitise; Pick et al. 2019), it seems so, but I did not find this information in the methods.

Line 169-172: how more likely? I would expect this topic to particularly suffer from publication bias, particularly depending on the method used to estimate inbreeding (see comment above). Did the authors test so? (see recommendations below).

Line 185-186: I recommend calling the analyses “Bayesian phylogenetic multilevel meta-regressions” to make it clear when effect sizes where weighted by the inverse of the sampling variance. I personally had to look at the code to understand this.

Line 187: please, provide the version of all R packages for reproducibility purposes (Pasquier et al. 2017).

Line 191-192: “1/(n-3)” instead?

Line 194-196: as far as I can see from the code, rather than the number of individuals in the population, what is used is the number of paired individuals. Wouldn't this number generally be different to the number of individuals used to estimate Z_r Average?

Line 205-206: shouldn't the wording rather be something like: “if evidence for inbreeding avoidance differ between species with and without inbreeding depression”? Also, “statistically significantly” (and throughout).

Line 215: Figure 3 indicates 39 data points rather than 38. Is that correct?

Line 243-244: from the code it seems that the authors randomly chose one model of the three, so I would state that here.

Line 298-299: I'm really not sure how appropriate this number is since it is a mean of means, and therefore uncertainty of the original means is neglected. It might be better to provide meta-analytic means here.

Line 311: what is "PM"? Apologies if I missed it.

Line 336-339: but the effect is not that far from statistical significance, so I would interpret these results with caution (as done for some other results shown below; more on this right below).

Line 373-376: I personally prefer this interpretation (i.e. prefer to avoid hard dichotomies), but the authors would need to adjust their interpretations of small but not statistically significant p-values throughout to be consistent - in this example CI's overlap with 0 and p-value is not < 0.05.

Line 380: please report the estimates or refer to the table that contains them.

Line 393-395: I would suggest to interpret these results more cautiously since only 7 data points were available for that specific regression, and despite non-statistical significance, the regression tends to be less steep than 1:1

Line 395-396: see my comments about considering this scenario when searching for studies (i.e. adding "preference" to the search string). Is Figure S2 correct here?

Line 408: I would suggest "might not be effective on its own". As suggested in the next sentence, I don't think we can be sure that if a mechanism is not reported for a species, that means it does not happen.

Line 429: the following reference seems relevant here (and throughout): Avilés and Purcell 2012

Line 451: I miss a paragraph discussing the potential limitations of the study (e.g. relatively few effect sizes, etc), what heterogeneity means for the interpretation, etc.

Line 461: "The data and code in this study"

Figure 1: I would recommend to add uncertainty for both axis to each dot (e.g. Figure 3 from Winney et al. 2018) so that the interpretation (e.g. line 325 and 335-336) is easier and more precise.

Figure 3: is the y-axis wording correct?

Table 2: consider showing residual variance for non meta-analytic models too.

Supplementary material

Minor: Olson et al. 2012 reference misses the last author, I believe.

Table S5: there seem to be duplicated rows.

Data

Please add the metadata provided in the supplementary material to the code and/or to an extra tab (or readme) in the excel file. Furthermore, please provide more bibliographic information (e.g. doi) in the dataset for the references included (and the excluded ones from Table S5) so that a reader can easily find them without having to match the supplementary material pdf with the dataset. All that would increase data reusability.

Overall, I would recommend a little bit of more data cleaning even if that will not change any results but just to increase standardization as much as possible. For example, remove unnecessary spaces in some levels (e.g. "cincta", "reticulata", see also variables "inbreeding_avoidance" and "Reported_in_study"), use the same format throughout (e.g. references, capital first letter for common_name), etc.

Cayuela et al., 2017: shouldn't the sample size be 50 rather than 60? Did the authors perform a data extraction double-check? If not, I would recommend to perform a double-check of a percentage of the data to confirm that data is generally correct (e.g. 25% as in Moran et al. 2020).

Code

The format of the dataset provided is .xlsx, whereas the code tries to import a .csv file. Please, confirm that the data provided is the correct one. I assumed so for my review, however, I got the following error for lines 47-49: "Error in `[data.frame](data, rh.idx, ids.var, drop = FALSE)` : undefined columns selected", which seems to contradict my assumption since the column "obs_or_expt" is missing.

As a general comment, I would recommend adding more inline comments to the code to make it easier to understand it. For example, I cannot easily follow the "weighted average" performed from line 41 on.

Code line 86-88: I believe this check shows that the `ott_id` for “*simochromis pleurospilus*” should be 474353 instead of 710012?

Code line 301: “`Method_of_IA2`” does not seem to be defined beforehand in the code, and as such, “`graph_1.b`” cannot be recreated. Please, revise accordingly.

Model M5 does not seem to converge nor run well, particularly the random effect part of the model, but also a little bit the fixed effect part. Plus, the effective sampling is very low, actually 0 for “units”. Did the authors had similar issues?

Overall, I would recommend the authors to save and provide the models they ran (e.g. save them as `.rda`) to increase reproducibility, and also to make it easier for researchers that want to run the code without having to wait for all the models.

Last, I feel a bit odd saying this and I might as well be totally wrong, if so, I wholeheartedly apologize, but I seem to recognize chunks of code as code I wrote for Sánchez-Tójar et al. 2020a (code: https://github.com/ASanchez-Tojar/meta-analysis_of_variance) and seemingly also Sánchez-Tójar et al. 2018 (code: https://github.com/ASanchez-Tojar/meta-analysis_sparrows_ssh)? If so, and since incentives are important in academia, I would appreciate if the authors would cite the original studies (Sánchez-Tójar et al. 2018, 2020a), the same way as we would normally do when reusing data or ideas from previous publications. If not, again, I apologize.

Sincerely, Alfredo Sánchez-Tójar. Note that I sign all my reviews since March 2018. I am more than happy to discuss any of the suggestions I have made with the authors and/or editors (my email is alfredo.tojar@gmail.com)

References:

- Avilés and Purcell 2012: <https://doi.org/10.1016/B978-0-12-394288-3.00003-4>
 Clark et al. 2019: <https://www.nature.com/articles/s41467-019-12283-6>
 Crnokrak and Roff 1999: <https://www.nature.com/articles/6885530>
 Dochtermann et al. 2019: <https://doi.org/10.1093/jhered/esz023>
 Duthie and Reid 2016: <https://doi.org/10.1086/688919>
 Haddaway et al. 2020: <https://t.co/QrGy3jjAuc?amp=1>
 Koricheva and Kulinskaya 2019: <https://doi.org/10.1016/j.tree.2019.05.006>
 Lange et al. 2017: <https://doi.org/10.1093/biolinnean/blw003>
 Leedale et al. 2020: <https://doi.org/10.1073/pnas.1918726117>
 Leroy 2014: <https://doi.org/10.1111/age.12178>
 Moran et al. 2020: <https://doi.org/10.1111/brv.12655>
 Nakagawa and Schielzeth 2010: <https://doi.org/10.1111/j.1469-185X.2010.00141.x>
 Nakagawa and Schielzeth 2013: <https://doi.org/10.1111/j.2041-210x.2012.00261.x>
 Nakagawa et al. 2020: <https://doi.org/10.1002/jrsm.1424>
 Pasquier et al. 2017: <https://doi.org/10.1038/sdata.2017.114>
 Pick et al. 2019: <https://doi.org/10.1111/2041-210X.13118>
 Sánchez-Tójar et al. 2018: <https://doi.org/10.7554/eLife.37385>
 Sánchez-Tójar et al. 2020a: <https://doi.org/10.1111/jeb.13661>
 Sánchez-Tójar et al. 2020b: <https://doi.org/10.1111/ele.13479>
 Senior et al. 2016: <https://doi.org/10.1002/ecy.1591>
 Thunken et al. 2011: <https://doi.org/10.1093/beheco/arq217>
 Wells et al. 2020: <https://doi.org/10.1111/ele.13578>
 Winney et al. 2018: <https://doi.org/10.1111/jeb.13197>
 Winternitz et al. 2016: <https://doi.org/10.1111/mec.13920>

Referee: 2

Comments to the Author(s)

This paper addresses a topic of great general interest to biologists; that is, the evolution of inbreeding avoidance mechanisms such as mating preferences for unrelated partners. My main concern is that the paper presents a simple linear causal relationship between inbreeding depression and inbreeding avoidance, where the former selects for the latter. I completely agree

that the severity of inbreeding depression is likely to be a strong selective force on inbreeding avoidance. However, the causal relationship between the two is likely to be more complex as patterns of inbreeding in the past also will determine the severity of inbreeding depression. For example, in species that have experienced frequent inbreeding in the past, there may have been purging of the rare, recessive and deleterious alleles that cause inbreeding depression. Thus, the association between inbreeding avoidance and inbreeding depression is likely to reflect a more complex causal relationship. I don't think this makes the results any less interesting, but I suggest that the authors revise some of their predictions (see specific comment #5) and some of the interpretation of their results (see specific comment #7).

Specific comments:

- (1) Lines 47–48: Please clarify what you mean by inbreeding? Inbreeding may refer to the process of mating in parental generation and/or it may refer to the production of inbred individuals in the offspring's generation. Selection for inbreeding avoidance would obviously happen at the parents' generation, whilst the fitness consequences are felt at the offspring's generation. There are also several definitions of inbreeding in the literature. For example, it may refer to mating between related individuals or non-random mating where related individuals are more likely to mate than expected by chance. For clarity, it might be useful if you state how you define inbreeding. Just to take an example where these definitions would matter: assume a small population consisting of a single male and a single female who were brother and sister. This is inbreeding according to a pedigree-based definition but not according to a definition based on random mating.
- (2) Line 69: Is kin recognition here the same as inbreeding avoidance by mate choice? Please clarify.
- (3) Lines 155–158: Is this your own metric for severity of inbreeding depression? If so, make this clear to the reader. If not, provide a reference for how it has been used in the past.
- (4) Lines 158–160: Please clarify whether the focal individual that is inbred or outbred corresponds with the individual whose traits you refer to here. For example, it is unclear here whether offspring mortality and offspring mass refers to the mass of inbred offspring or the offspring of inbred parents?
- (5) Lines 201–205: I'm not convinced by this argument. This relationship may also be less than one if the absence of inbreeding avoidance has led to frequent inbreeding in the past and thereby purging of rare, deleterious and recessive alleles. See also major comment.
- (6) Lines 299–301: Please check this statement. It is unclear to me how relatedness can be negative if it is bounded between 0 and 1 (see lines 130–133)?
- (7) 339–343: An alternative explanation here is that the lack of inbreeding avoidance has been associated inbreeding in the past, which has purged the population of the rare, deleterious and recessive alleles that cause inbreeding depression. See also major comment.
- (8) Line 372: Please check this subheading and similar statements throughout. I appreciate that this is not your intention, but this might be read as a 'good for the species' argument.
- (9) Lines 380–382: I'm not sure if I get this argument. Even if a given species has evolved inbreeding avoidance mechanisms, it seems unlikely that such mechanisms would be perfect. If not, there should be some risk of inbreeding and if this is the case, should we not expect inbreeding depression to be more severe when parents are more related? Is this finding robust or does it reflect low statistical power due to few instances of inbreeding between closely related parents?
- (10) Lines 395–396: Consider rewording this statement. I would argue that there is always some transmission benefit (or inclusive fitness benefit) of inbreeding as it increases the probability that a given gene is transmitted. However, this benefit is often outweighed by the cost due to the reduction in fitness of inbred offspring. Presumably, this statement refers to the net benefit of inbreeding?
- (11) Line 399: Individuals avoid inbreeding (not species).
- (12) Lines 439–440: Please add a reference to support this claim.

Per Smiseth (I always sign my reviews)

Author's Response to Decision Letter for (RSPB-2020-2372.R0)

See Appendix A.

RSPB-2021-1045.R0

Review form: Reviewer 1 (Alfredo Sánchez-Tójar)

Recommendation

Major revision is needed (please make suggestions in comments)

Scientific importance: Is the manuscript an original and important contribution to its field?

Good

General interest: Is the paper of sufficient general interest?

Good

Quality of the paper: Is the overall quality of the paper suitable?

Acceptable

Is the length of the paper justified?

Yes

Should the paper be seen by a specialist statistical reviewer?

No

Do you have any concerns about statistical analyses in this paper? If so, please specify them explicitly in your report.

Yes

It is a condition of publication that authors make their supporting data, code and materials available - either as supplementary material or hosted in an external repository. Please rate, if applicable, the supporting data on the following criteria.

Is it accessible?

No

Is it clear?

Yes

Is it adequate?

N/A

Do you have any ethical concerns with this paper?

No

Comments to the Author

I have now carefully read the authors' responses to the reviewers, and the new version of the manuscript. I am very satisfied with the changes implemented. The authors have made an excellent and thorough job dealing with the concerns of the reviewers. Thank you very much. I'm convinced that the study has consequently improved. I have only a few more comments and suggestions left, mostly minor, but two potentially major ones, that I recommend dealing with.

Some models seem to be over-fitted, so I would suggest either to not present them or to explicitly and very clearly warn the reader that their results are unlikely to be reliable. The most extreme case seems to be model 5 (line 197), which, if I understood correctly, estimates 18 fixed effect estimates plus 1-2 random effect estimates. That is, it estimates about 20 estimates, but it only includes 34 data points (is this interpretation correct?). Other models that seem to be in high risk of being over-fitted are models 3, 6, 7, and 8. As a rule of thumb, I normally aim at 10 data points per model estimate. The minimum I have seen recommended is 4 data points per model estimate, which I personally consider extremely low. Related to this, I would recommend to remind the reader in the discussion section that unfortunately only very few studies (41 max) were available, and that we need many more studies on this topic to obtain a clearer and more precise picture about the hypotheses tested.

An I² of 21% (line 220) is an unusually low level of heterogeneity. I would recommend the authors to discuss what this means for their results in the discussion section (see Rutkowska et al. 2014 as example). In short, that I² suggest that most (79%) of the differences observed in results across studies are simply explained by differences in sampling variance (i.e. sample size) between studies and only 21% are potentially due to methodological and/or biological effects. For meta-analyses in ecology and evolution, normally only ~5% of the differences in results observed across studies are explained by differences in sampling variance (Senior et al. 2016), making this result, in my opinion, remarkable and worth interpreting and discussing in detail.

Minor:

Line 207-210: I'm unsure about adding a predictor's SE to the model. As far as I can see, it simply tests whether the uncertainty of the predictor is related to the response variable, which I don't think is what the authors intended to do when adding this predictor? Could the authors expand on this?

Line 237-242: I did not know it was possible to run a binary meta-analysis in MCMCglmm. For the weights, I wonder whether it might make sense to weight by $1/(n-3)$ just to keep consistency with before, even though the response is not a Zr.

Line 286: where effect sizes weighted by the inverse of the sampling variance in the Egger's tests too? They should, in principle, be. Same for the time-lag bias test.

Line 303-304: I would suggest adding something like "although the effect was small and uncertain" here, and to show here the heterogeneity estimates.

Line 315: I would recommend adding how many out of how many species to be more explicit here; e.g. "In most species where inbreeding depression has been reported (X out of X),...". Same in line 326.

Line 450-451: I find it a bit confusing that all of the sudden female fig wasps are introduced here. I had to re-read the paragraph a couple of times.

Line 466: I tried to access the data using this doi, but could not. I'm guessing the link will be active after acceptance.

Figure 3: the authors could present the 95% CrI rather than SE to keep consistency and make it more easily interpretable.

Table 2: is the pMCMC of model 2 difference from 1 correct? I recommend reporting the random effect and residual variance estimates for each model either here or in the supplements.

Supplementary material: if possible, I would include the tables in the file "Supplementary Info.pdf", it was a bit cumbersome to navigate the tabs and adjust column size in the excel file.

SM Line 77: "no statistically significant interaction"?

SM Line 143-144: in this case it could also be driven by the very small number of effect sizes available for the pedigree estimate (n=6), so I would highlight this potential cause as done in some of the analyses above.

Edits/typos:

Line 67: "sex-biased" (line 366)

Line 242: I guess Nspecies = 34 could be deleted since that information is also provided in the following sentence.

Line 358: "was"?

Sincerely, Alfredo Sánchez-Tójar. Note that I sign all my reviews

References:

Nakagawa et al. 2021: <https://doi.org/10.32942/osf.io/k7pmz>

Senior et al. 2016: <https://www.jstor.org/stable/44082191>

Rutkowska et al. 2014: <https://doi.org/10.1111/jeb.12282>

Review form: Reviewer 2

Recommendation

Accept as is

Scientific importance: Is the manuscript an original and important contribution to its field?

Good

General interest: Is the paper of sufficient general interest?

Excellent

Quality of the paper: Is the overall quality of the paper suitable?

Excellent

Is the length of the paper justified?

Yes

Should the paper be seen by a specialist statistical reviewer?

No

Do you have any concerns about statistical analyses in this paper? If so, please specify them explicitly in your report.

No

It is a condition of publication that authors make their supporting data, code and materials available - either as supplementary material or hosted in an external repository. Please rate, if applicable, the supporting data on the following criteria.

Is it accessible?

N/A

Is it clear?

N/A

Is it adequate?

N/A

Do you have any ethical concerns with this paper?

No

Comments to the Author

The authors have address all my comments on the previous version of the manuscript and they an excellent job in revising this manuscript. This manuscript would in my opinion make a valuable contribution to the field.

Per Smiseth (I sign all my reviews)

Decision letter (RSPB-2021-1045.R0)

21-Jun-2021

Dear Miss Pike:

Your manuscript has now been peer reviewed and the reviews have been assessed by an Associate Editor. The reviewers' comments (not including confidential comments to the Editor) and the comments from the Associate Editor are included at the end of this email for your reference. As you will see, the reviewers and the Editors have raised some concerns with your manuscript and we would like to invite you to revise your manuscript to address them.

Research ethics:

Use of animals and field studies:

If your study uses animals please include details in the methods section of any approval and licences given to carry out the study and include full details of how animal welfare standards were ensured. Field studies should be conducted in accordance with local legislation; please

include details of the appropriate permission and licences that you obtained to carry out the field work.

It is a condition of publication that you make available the data and research materials supporting the results in the article (<https://royalsociety.org/journals/authors/author-guidelines/#data>). Datasets should be deposited in an appropriate publicly available repository and details of the associated accession number, link or DOI to the datasets must be included in the Data Accessibility section of the article (<https://royalsociety.org/journals/ethics-policies/data-sharing-mining/>). Reference(s) to datasets should also be included in the reference list of the article with DOIs (where available).

Please submit a copy of your revised paper within three weeks. If we do not hear from you within this time your manuscript will be rejected. If you are unable to meet this deadline please let us know as soon as possible, as we may be able to grant a short extension.

Best wishes,
Dr Sasha Dall
mailto: proceedingsb@royalsociety.org

Associate Editor

Comments to Author:

The authors should discuss potential problems that might occur with overfitting the models.

Reviewer(s)' Comments to Author:

Referee: 2

Comments to the Author(s).

The authors have address all my comments on the previous version of the manuscript and they an excellent job in revising this manuscript. This manuscript would in my opinion make a valuable contribution to the field.

Per Smiseth (I sign all my reviews)

Referee: 1

Comments to the Author(s).

I have now carefully read the authors' responses to the reviewers, and the new version of the manuscript. I am very satisfied with the changes implemented. The authors have made an excellent and thorough job dealing with the concerns of the reviewers. Thank you very much. I'm convinced that the study has consequently improved. I have only a few more comments and suggestions left, mostly minor, but two potentially major ones, that I recommend dealing with.

Some models seem to be over-fitted, so I would suggest either to not present them or to explicitly and very clearly warn the reader that their results are unlikely to be reliable. The most extreme case seems to be model 5 (line 197), which, if I understood correctly, estimates 18 fixed effect estimates plus 1-2 random effect estimates. That is, it estimates about 20 estimates, but it only includes 34 data points (is this interpretation correct?). Other models that seem to be in high risk of being over-fitted are models 3, 6, 7, and 8. As a rule of thumb, I normally aim at 10 data points per model estimate. The minimum I have seen recommended is 4 data points per model estimate, which I personally consider extremely low. Related to this, I would recommend to remind the reader in the discussion section that unfortunately only very few studies (41 max) were available, and that we need many more studies on this topic to obtain a clearer and more precise picture about the hypotheses tested.

An I2 of 21% (line 220) is an unusually low level of heterogeneity. I would recommend the authors to discuss what this means for their results in the discussion section (see Rutkowska et al. 2014 as example). In short, that I2 suggest that most (79%) of the differences observed in results across studies are simply explained by differences in sampling variance (i.e. sample size) between studies and only 21% are potentially due to methodological and/or biological effects. For meta-analyses in ecology and evolution, normally only ~5% of the differences in results observed across studies are explained by differences in sampling variance (Senior et al. 2016), making this result, in my opinion, remarkable and worth interpreting and discussing in detail.

Minor:

Line 207-210: I'm unsure about adding a predictor's SE to the model. As far as I can see, it simply tests whether the uncertainty of the predictor is related to the response variable, which I don't think is what the authors intended to do when adding this predictor? Could the authors expand on this?

Line 237-242: I did not know it was possible to run a binary meta-analysis in MCMCglmm. For the weights, I wonder whether it might make sense to weight by $1/(n-3)$ just to keep consistency with before, even though the response is not a Zr.

Line 286: where effect sizes weighted by the inverse of the sampling variance in the Egger's tests too? They should, in principle, be. Same for the time-lag bias test.

Line 303-304: I would suggest adding something like "although the effect was small and uncertain" here, and to show here the heterogeneity estimates.

Line 315: I would recommend adding how many out of how many species to be more explicit here; e.g. "In most species where inbreeding depression has been reported (X out of X),...". Same in line 326.

Line 450-451: I find it a bit confusing that all of the sudden female fig wasps are introduced here. I had to re-read the paragraph a couple of times.

Line 466: I tried to access the data using this doi, but could not. I'm guessing the link will be active after acceptance.

Figure 3: the authors could present the 95% CrI rather than SE to keep consistency and make it more easily interpretable.

Table 2: is the pMCMC of model 2 difference from 1 correct? I recommend reporting the random effect and residual variance estimates for each model either here or in the supplements.

Supplementary material: if possible, I would include the tables in the file "Supplementary Info.pdf", it was a bit cumbersome to navigate the tabs and adjust column size in the excel file.

SM Line 77: "no statistically significant interaction"?

SM Line 143-144: in this case it could also be driven by the very small number of effect sizes available for the pedigree estimate (n=6), so I would highlight this potential cause as done in some of the analyses above.

Edits/typos:

Line 67: "sex-biased" (line 366)

Line 242: I guess Nspecies = 34 could be deleted since that information is also provided in the following sentence.

Line 358: "was"?

Sincerely, Alfredo Sánchez-Tójar. Note that I sign all my reviews

References:

Nakagawa et al. 2021: <https://doi.org/10.32942/osf.io/k7pmz>

Senior et al. 2016: <https://www.jstor.org/stable/44082191>

Rutkowska et al. 2014: <https://doi.org/10.1111/jeb.12282>

Author's Response to Decision Letter for (RSPB-2021-1045.R0)

See Appendix B.

Decision letter (RSPB-2021-1045.R1)

12-Jul-2021

Dear Miss Pike

I am pleased to inform you that your manuscript entitled "Why don't all animals avoid inbreeding?" has been accepted for publication in Proceedings B.

Data Accessibility section

Open Access

Paper charges

Sincerely,

Dr Sasha Dall

Associate Editor:

Board Member

Comments to Author:

The authors made a great job in dealing with the referees comments.

Appendix A

Department of Zoology
University of Oxford
Zoology Research and Administration Building
11a Mansfield Road
Oxford
OX1 3SZ
Email: victoria.pike@zoo.ox.ac.uk

Dear Dr Dall and Professor Barrett,

Please find attached the resubmission of our manuscript RSPB-2020-2372 entitled “Why don’t all animals avoid inbreeding?”.

We thank the reviewers for the time and effort they invested to provide us with extremely useful and constructive comments. This has allowed us to greatly improve our manuscript and clarify crucial details. An altogether positive review experience – thank you!

We have now addressed the key requests to include more detail in the manuscript by re-writing the methods and incorporated new analyses. We have performed additional literature searches prompted by the referee’s comments and applied more stringent exclusion criteria. All analyses have been re-run with the updated datasets and the conclusions of our paper remain unchanged.

Please find below a point-by-point response to the reviewers’ comments.

Yours sincerely,

Victoria Pike

Response to Reviewer 1 (reviewers comments bolded, our responses normal text, line numbers refer to the manuscript without tracked changes)

The authors performed a meta analysis and investigated inbreeding avoidance across the animal kingdom. The topic is very interesting and I enjoyed reading the manuscript.

At the same time I feel that the methodology part is missing some cubical information, i.e. how was the literature used in the meta analysis found, i.e. exact search string as pointed out by one of the referees.

We have supplemented our methods with the additional information requested (see below).

In this study, the authors used a meta-analytic approach to test multiple hypotheses about inbreeding avoidance, including under which circumstances inbreeding avoidance is expected to preferentially evolve. To do so, they compiled a dataset containing 48 effect sizes (47 animal species). The manuscript is well written and the hypotheses are very interesting, however I have some major methodological concerns about the literature search, the data extraction and the statistical analyses. My main concerns and suggestions are described below followed by additional line-to-line comments. I apologize for the length of my review, but I hope my comments will be helpful in revising this contribution.

No need to apologise – thank you! We are grateful for the time invested by the reviewers to provide us with detailed, constructive feedback.

Literature search

My main concerns about the literature search are: (1) the lack of reporting essential information to understand what was done and to allow reproducibility; and (2) the potential incompleteness and, hopefully to a lesser extent, bias of the search strategy used.

We have improved the reporting of our search protocols and indicate steps taken to avoid missing and biased data – details provided in response to specific points below.

(1) It is unclear what specific search string was used and which databases within Web of Science (WoS) were searched. Presumably the authors searched WoS Core Collection and did a topic (TS) search (title, abstract, keywords), which depending on the authors' subscription would cover more or less databases (see "Citation Indexes" in tab "Advanced Search").

Regarding the search string, I would assume the authors searched for "(inbreeding avoidance OR incest avoidance)" rather than "(inbreeding avoidance AND incest avoidance)". However, the reported "N ~ 1,400" in Figure S1 would indicate that the authors indeed searched for studies containing both keywords "inbreeding avoidance AND incest avoidance" - my own searching of studies containing "inbreeding avoidance"

OR "incest avoidance" led to 1381 hits from WoS Core Collection and 13,200 hits from Google Scholar.

The searches we ran did include separate searches on both “inbreeding avoidance” and “incest avoidance”. We have now clarified the string searches we used in our analysis in the methods (lines 106 to 109):

- i) ‘inbreeding avoidance’ (n = 1266)
- ii) ‘incest avoidance’ (n = 228)
- iii) ‘inbreeding preference’ (n = 12)
- iv) ‘incest preference’ (n = 0)

In addition, we have now included a table (Table S25) including all the search results for each of these searches.

Additionally, Figure S1 would require some corrections. I don’t think it is good practice to provide approximate values because the purpose of the PRISMA diagram is to increase the reproducibility of the literature search, and it’s unclear how approximated values would help with that. Therefore, I would lean towards stating that those values are unknown, if that is the case. Furthermore, numbers do not seem to add up for the fulltext screening and included steps (e.g. number of papers screened and excluded is the same).

We have now removed the approximate values and replaced them with exact values (see Figure S1). In addition, we have included a supplementary data with the search results to increase reproducibility (see Table S25 & S26).

Last, did the authors screened ~27000 titles and abstracts manually or used automatic procedures such as machine learning? Overall, although not always reported in evidence synthesis in ecology and evolution, all these details are important to increase reproducibility in evidence synthesis (e.g. Sánchez-Tójar et al. 2020b; Haddaway et al. 2020), so I would suggest to provide as many as possible and acknowledge those that cannot be provided as limitations.

We have clarified in our methods that we used a manual approach to screen the studies (see line 109-110) and have included a supplementary table (Table S25) with details of the references we screened.

(2) Assuming the authors searched for (“inbreeding avoidance” OR “incest avoidance”), this strategy would seemingly miss presumably important alternative keywords such as (“avoid* inbreeding” OR “avoid* incest”) (and potentially more combinations). Indeed, simply adding (“avoid* inbreeding” OR “avoid* incest”) to the search string leads to 15% increase in hits in WoS Core Collection (200+ references) and potentially hundreds more hits from Google Scholar. In addition, using (“inbreeding avoidance” AND “incest avoidance”) could be biased towards finding only papers that found evidence for avoidance but not for preference (or no association). My suggestion would therefore be to also include “preference” in the search string, e.g. (“inbreeding preference” OR “incest preference”), which will find a few extra hits that could be important (e.g. Thunken et al.

2011, Lange et al. 2017). The two potential problems highlighted in point (2) might have been mitigated thanks to the forward and backward searches performed by the authors, however, whether that was the case would require confirmation in order to evaluate the quality of the search strategy.

Thank you for this suggestion, as the reviewer highlights above, we did mitigate these problems by conducting forward and backward searches on the studies in our analysis. However, we conducted some additional searches in response to this suggestion (see lines 106 to 109):

- i) “inbreeding preference”
- ii) “incest preference”

We did not find any papers that were not already included in the analysis, which fully met our inclusion criteria. We came across the two papers mentioned above but they did not meet our inclusion criteria:

- iii) Thünken et al 2011 – This study did not directly examine mate choice in relation to relatedness, but investigated inbreeding preference based on olfactory cues in cichlids (*Pelvicachromis taeniatus*) (see table S24).
- iv) Lange et al 2017 – This study was about the effects of inbreeding on various female traits in the West African cichlid (*Pelvicachromis taeniatus*) and thus it did not investigate the relatedness between mating pairs. In addition, some of the study was conducted on a captive population (not included in table S24 as this study was excluded prior to the full text search).

We did find one study that met our inclusion criteria which was published after our original searches were carried out. This study was on the coppery titi monkey (*Plecturocebus cupreus*, (Dolotovskaya et al. 2020)) and we have included this in our analysis. In addition, there were other examples of species with inbreeding preferences, such as the social lizard, *Liopholis whitii* (Bordogna et al. 2016), retrieved by our new searches. This particular study was not included in our meta-analysis as it did not meet our inclusion criteria, but we reference it in the introduction, and it is listed in table S24.

Data extraction

My main concerns about data extraction are whether: (1) average relatedness between potential mates is comparable across studies; and (2) the effect size chosen in the correct one. I do not have a definite answer about these two points but would like to hear the authors’ thoughts. Also, I’m not an expert on inbreeding avoidance, so I apologize in advance for any misunderstanding.

Our aim is to make our paper easy to follow for interested readers whether they have expertise in inbreeding avoidance or not, so we thank the reviewer here for pointing out where there are ambiguities (see our responses below).

(1) I don't understand very well what the "average relatedness between potential mates in the population" represents across studies. I tried to understand it better by looking at three included studies but there seem to be some conceptual differences in how that average relatedness was calculated across them, and I wonder whether those differences should be taken into account in the analyses. For example, the values extracted from Barati et al. (2018) seem to correspond to genetic relatedness for breeding pairs (ca. 0.05) and for breeding females and their helpers (ca. 0.18; Figure 1), whereas the values extracted from Cayuela et al. (2017) seem to correspond to genetic relatedness for males and females from the same (0.21) and different breeding patches (0.04); and the values extracted from Griffin et al. (2003) seem to correspond to genetic relatedness for breeding pairs (ca. 0.05) and all pairs in the population (0.23; Figure 5). A priori, I was expecting that the average relatedness between potential mates in the population would be always calculated as in the latter example (Griffin et al. 2003), and I wonder whether the other strategies are indeed comparable. Perhaps the author could explain this further.

To clarify, the average relatedness in the population refers to the average relatedness between males and females in the study population (see lines 125-128). Our intention is to use a metric that captures average relatedness between pairs across a range of studies and species. While this has the advantage of allowing us to include a wider range of species, it does lead to difficult cases such as the noisy miner (*Manorina melanocephala*) that has a complex social structure where related and unrelated birds help to care for broods (Barati et al. 2018).

Consequently, and in light of these comments, we have now made our inclusion criteria for estimating average relatedness in populations more stringent. We have now checked all studies included in the analyses against these new criteria and removed any cases where relatedness between males and females in the study population may be affected by other factors e.g. increased relatedness due to including helpers at the nest. With this stricter inclusion criteria, we have removed the following seven studies (listed in table S24 along with the reasons for their exclusion):

- 1) (Barati et al. 2018) – the noisy miner (*Manorina melanocephala*)
- 2) (Brekke et al. 2012) – the hihi (*Notiomystis cincta*)
- 3) (Cayuela et al. 2017) – the yellow-bellied toad (*Bombina variegata*)
- 4) (Van Dijk et al. 2015) – the sociable weaver (*Philetairus socius*)
- 5) (Varian-Ramos and Webster 2012) – the red-backed fairy-wren (*Malurus melanocephalus*)
- 6) (Leedale et al. 2018) – the long tailed tit (*Aegithalos caudatus*)
- 7) (Billing et al. 2012) – the house sparrow (*Passer domesticus*)

(2) Can estimates of relatedness really be considered correlation coefficients rather than mean proportions/percentages of similarity across pairs? I have doubts whether it is conceptually valid to treat those means as correlation coefficients, so perhaps the authors could explain this further. I wonder whether for the main purposes of this meta-analysis, calculating a mean ratio/difference (e.g. lnRR or SMDH) between mean relatedness

between pairs and average relatedness would be a more appropriate effect size. This would also allow the authors to accommodate the differences in uncertainty (e.g. SE) between estimates of Zr_{Paris} and Zr_{Average} , and to take into account the sampling variance of Zr_{Average} , which so far it is only approximately taken into account by using $\log(n)$? (see comment about $\log(n)$ below). If the authors decide to treat them as correlations, they should be aware that, due to the [0,1] bounding, Fisher's z-transformations will not help much with normalizing the distribution of the data, and so, it might be a good idea to run sensitivity analyses using beta regressions (as done in Dochtermann et al. 2019).

Relatedness in this context is a correlation coefficient of genetic similarity between pairs of individuals relative to the reference population (Grafen 1985; Wright 1992; Wang 2017). Theoretically, it can range from 1 (clones) to -1 (minus values occur when two individuals are less related than the population average).

Using estimates of relatedness, rather than calculating an effect size such as $\ln RR$ or SMDH, is crucial to testing the main prediction of our paper: inbreeding avoidance only evolves when relatives interact. If related males and females never encounter each other then selection for inbreeding avoidance will not occur. Therefore, it is only when relatedness in the population (r_{Average}) is relatively high that we expect inbreeding avoidance (relatively low values of r_{Pairs}) to occur. When r_{Average} is low even random mating will result in low r_{Pairs} .

If we use r_{Pairs} and r_{Average} to calculate an effect size then we cannot test this main prediction. An effect size would examine if there is inbreeding avoidance or not, irrespective of the frequency with which relatives interact. This can be illustrated using two hypothetical examples, one where $r_{\text{Pairs}} = 0.1$ and $r_{\text{Average}} = 0.1$ and another where $r_{\text{Pairs}} = 0.5$ and $r_{\text{Average}} = 0.5$. The effect size will be the same for both examples, but in the first scenarios relatives rarely interact so selection for inbreeding avoidance will always be weak, whereas in the second scenario relatives frequently interact and there is the potential for selection for inbreeding avoidance.

It is the effect of the frequency with which related individuals interact within populations that is missing from many studies of inbreeding, which motivated our work. We therefore believe the approach we use is robust and makes the effect of average relatedness in populations explicit.

Regarding data transformation, given that relatedness is a correlation coefficient that can range from -1 to 1 we believe Fisher's z-transformation is in principle appropriate. The values for relatedness in our actual data ranged from -0.07 to 0.23 for pairs and -0.12 to 0.30 for populations and were approximately normal. As values are negative, beta regressions cannot be used.

Statistical analyses

My main concerns about the statistical analyses are: (1) the calculation of I2 for meta-regressions instead of R2marginal; and (2) the validity of the publication bias tests performed.

Thank you very much for the suggestions and pushing us to clarify our methodological approaches. Please find our detailed responses below.

(1) It is not very intuitive to calculate – or better said, to call it, “heterogeneity I2” in meta-regressions. I2 is a relative measure of heterogeneity that is normally calculated for meta-analytic (i.e. intercept-only) models to show whether heterogeneity is present (and how much and from where it comes from), and thus, to inform us whether meta-regressions should be implemented to try and explain heterogeneity, if any found (Senior et al. 2016). Therefore, it is uncommon to estimate I2 from meta-regressions, but R2marginal instead (Nakagawa and Schielzeth 2013), because the latter gives us an approximation of the percentage of heterogeneity observed in the meta-analytic (intercept-only) model explained by the moderator(s) included in the subsequent meta-regression. Estimating I2 from a meta-regression would be equivalent to estimating adjusted repeatability (Nakagawa and Schielzeth 2010), and as such, we should consider whether the variance explained by the moderators (R2marginal) needs to be added to the denominator when estimating I2, which depends on what we want I2 to reflect (Nakagawa and Schielzeth 2010). Overall, my preference would be to calculate and discuss R2marginal for all meta-regressions (see code from Sánchez-Tójar et al. 2018), and to simply report the non-standardized variance components from the meta-regressions; although the authors could consider providing H2 (i.e. I2phylogeny; sensu Nakagawa and Santos 2012) and discuss the biological importance of those results.

We are grateful to the referee for pointing out that this was confusing. Our intention was to give a breakdown of the variance explained by our different random effects.

As suggested by the referee we now provide an intercept only model (M1 in the R script, Table S2, lines 213-220) where we estimate total heterogeneity (phylogenetic variance + residual variance + sampling variance) and phylogenetic heritability (phylogenetic variance / sum(phylogenetic & residual variance)). This shows there is a large amount of unexplained variance, paving the way for our subsequent analyses. For all other meta-regression models, we provide R2 marginal values (see Tables 2 and S2-S20. We have now clarified our approach in the methods section.

(2) Regarding the publication biases tests, I don't understand well what was done, particularly since the code for these analyses was seemingly not provided. From the description and the plots presented (Figures S8-S10), I suspect the publication bias analyses might not have been conducted appropriately. As far as I know, regtest() and trimfill() are not yet implemented for multilevel models. Also, I don't understand why the x-axis of Figures S8-S10 says log odds ratio and not Zr (i.e. Fisher's Z), and I don't seem to find the trim and fill numeric results anywhere. To properly test for publication bias in multilevel models, the random effects variance should to be taken into account either by extracting meta-analytic residuals (more in Nakagawa and Santos 2012; see code from Sánchez-Tójar et al. 2018 – depending on the version of MCMCglmm used) or perhaps

easier making use of multilevel meta-regressions that include precision (i.e. the inverse of the sampling variance) as a moderator. That would correspond with an adjusted Egger's test that would show whether effect sizes approximate zero with an increase in precision (see example in section 5.6 from supplementary material in Sánchez-Tójar et al. 2020b). I would also recommend the authors to test for time-lag bias by including "year of publication" as moderator (e.g. Sánchez-Tójar et al. 2018; reviewed in Koricheva and Kulinskaya 2019). Last, as for the trim and fill method, I would actually suggest not running it as this method does not seem to work very well when heterogeneity is present, and overall its results can be misleading since it is not possible to know if and how many effect sizes remained unpublished.

Thank you for the suggestions and we agree with all the above points. In the revised version we have now provided the following:

- 1) Modified Egger's test using a multilevel meta-regression model including the sampling variance as an explanatory variable following the referees advice that is also outlined in (Sánchez-Tójar et al. 2020). We also included *ZrAverage* in this analysis as we expect publication bias to influence the residuals of the relationship between *ZrPairs* and *ZrAverage* (lines 283-295 and PB1 in the R script).
- 2) Tested for time lag effects using a meta-regression with year as an explanatory variable (see lines 292-295 PB2 in the R script).
- 3) Revised our funnel plots using meta-analytic residual from our intercept only model (see Figures S2&3).
- 4) Removed the trim and fill analyses
- 5) Added tests of publication bias and time lag effects in *ZrDepression* following the same approach as in 1 & 2 (PB3 & PB4 in the R script).

All the code mentioned above is included in the supplementary code file (uploaded to dryad: <https://datadryad.org/stash/share/MQCP6J9rPsdIxeUDJfStB-t6jn2aft2s8G2SkvEVEr0>, doi:10.5061/dryad.6hdr7sr0t).

PS: Please, consider citing all the references of the included studies in the main text so that primary researchers get the credit they deserve. If the limitation is how many references Proceedings B allows to be cited, I would like to take advantage and ask Proceedings B to allow the authors to cite all references included in their analyses. Other journals allow so (e.g. Biological Reviews, Nature Ecology and Evolution) and there is no clear reason why Proceedings B should not adopt this when publishing evidence synthesis.

We tried to cite all the studies included in the analysis, but this caused the page limit to be exceeded at resubmission. We would be happy to include these references if our page limit was increased. We have included the full references in Table S1 and included a full list of references in the supplementary materials (see page 27 of Supplementary Materials).

Line-to-line comments

Line 60-61: I wonder whether MHC-dependent mate selection would be a specific type of

inbreeding avoidance/preference worth integrating (meta-analysis by Winternitz et al. 2016).

This is an interesting point. Unfortunately, in most studies the mechanism by which species distinguish between relatives and non-relatives was not investigated pushing this beyond the scope of our study.

Line 74: probably also highly variable across populations of the same species?

Yes. We have now noted that 'inbreeding depression can be highly variable both within and across species' (see line 80-82).

Line 84: I would use "phylogenetic meta-analysis" here and throughout (e.g. the abstract) to use consistent terminology and make it easier for the reader.

We have now edited the manuscript and used 'phylogenetic meta-analysis' throughout the main text and abstract (see lines 33 & 89-90).

Line 93-94: something that was not explicitly estimated was that variation in the strength. I think that before running the meta-regressions and testing all these hypotheses, the authors could run a Bayesian phylogenetic multilevel meta-analysis (i.e. intercept-only model) to show what is the overall effect size for *ZrPairs* (although see effect size suggestion), it's 95% Credible Intervals, it's 95% Prediction intervals (more in Nakagawa et al. 2020; Sánchez-Tójar et al. 2020b), and the absolute (Q test) and relative (I²) heterogeneity. Then, the meta-regressions would aim at explaining the heterogeneity found in the intercept-only model. Just a suggestion that depends also on the final effect size of choice.

We have now added a model (M2 in the R script, lines 220-222) that examines the overall effect size and added estimates from this model to the main text (line 306). In addition, we have added an orchid plot of *ZrPairs* for species with and without inbreeding depression so that 95% confidence intervals, 95% prediction intervals and the precision of estimates can be visualised (see Figure S5).

Line 104-105: if possible, please, provide the full list of studies found by the searches to increase reproducibility.

We have provided a supplementary table with the full list of studies found by the searches (Table S25). In addition, an updated list of studies has been provided where we screened the full text (290 studies listed in the supplementary material (Tables S24 & 26)) and our PRISMA flow chart has been updated (see Figure S1) with the details of our searches for reproducibility.

Line 105-106: does "all" refer to the 287 studies included in fulltext screening? Could you please provide in the supplements the list of studies for which forward and backward searches were performed? Was this process done manually or automatically? Also, do I understand correctly that only studies on wild populations were included in this meta-

analysis? If so, I would expect to see that criterion appear in the list of exclusion reasons provided in Table S5.

Yes, the forward and backward searches were performed on all of the studies included in the full text screening, we have clarified this in the main text (see lines 112-114). Studies on captive populations were excluded from our analysis, which we have rephrased in the methods for clarity (see lines 110-112). This reason for exclusion does not frequently appear in Table S24 (although it does appear for (Bolton et al. 2012; Mishra et al. 2020; Rabier et al. 2020; Carleial et al. 2020)) as studies on captive populations were often excluded at an earlier stage as it was clear from reading the titles and/or abstract that the study was conducted on a captive species.

Line 113-134: since different approaches to estimate relatedness (pedigree vs. markers vs. genome-wide) can lead to more or less precise estimates, I would consider testing if estimates differ depending on the method used (assuming there were more than one method use across studies). Additionally, estimates of relatedness can also be more or less precise depending on the quality of, for example, the pedigree, so I would consider exploring whether those potential methodological differences might explain some of the variance in results across studies. That is, effect sizes here likely vary depending on both sampling variance (i.e. number of individuals) and genetic information to reconstruct relatedness.

We have now carried out this recommended analysis, (Model S7) examining if estimates of inbreeding avoidance depend on how relatedness was measured. We found that for the two different types of relatedness measurements (pedigree vs genetic markers) there was no difference in the estimates of inbreeding avoidance (see lines 56-59 and 137-145 of supplementary material and Table S20 & S22).

Line 122-124: as far as I can see in the data, sample sizes of both populations are the same (N=270; is that correct?) meaning that a weighted and non-weighted mean would be in principle the same?

Yes, this is correct we have edited the text to 'average' rather than 'weighted average' (see line 132-134).

Line 141-142: did the authors perform additional searches to find more information about these mechanisms from other studies, just to make sure there is agreement across the literature for each species? Does this include only papers included in the final analysis (n = 48) or all the others (n=287; Table S5)? I guess there could be important information for these classifications in all papers listed in Table S5.

All of the mechanisms listed were based on the authors definition in each of the studies. We have clarified this in lines 152 to 165. In any cases where we used data from additional studies this was noted in Table S1, and only occurred when investigating inbreeding depression in the supplementary analysis (see column 'Measurement of ID used to calculate r'). Only in three cases, was data used that were not from the original inbreeding study, but

were from a study referenced as evidence of inbreeding depression or a lack of inbreeding depression in that species:

- 1) (Dunn et al. 2012) who cite their previous study as evidence of inbreeding depression in pronghorns (Dunn et al. 2011)
- 2) (Robinson et al. 2012) who cite their previous study as evidence of inbreeding depression in fruit flies (Robinson et al. 2009)
- 3) (Collet et al. 2020) who cites (Vayssade et al. 2014) as evidence of inbreeding depression in the parasitoid wasp

We have added this detail into the methods (see line 166-179) and cited the papers above in the main text.

Line 148: doesn't this assume that extra-pair copulations' are a strategy of inbreeding avoidance even when this might not be necessarily true across species?

We edited this sentence to acknowledge that extra-pair copulations are not always a mechanism of avoiding inbreeding and cite (Duthie et al. 2016) (see line 160-161). However, in the context of the studies we included in the analysis extra-pair copulations were treated as a mechanism of inbreeding avoidance when the authors stated this.

Like 149-150: this doesn't sound optimal; shouldn't they instead be classified as "NA" or perhaps "unclear" to still be able to use the data?

We have now edited this sentence for clarification (see lines 162-163 and Table 1).

Line 154: when discussing these results, it might be interesting to compare them with that of previous meta-analyses on this topic: Crnokrak and Roff 1999 (wild organisms), Leroy 2014 (livestock), and Clark et al. 2019 (humans)

Thank you for highlighting these studies, we have now cited two of these papers in the manuscript (Crnokrak and Roff 1999; Leroy 2014) (see lines 51, 176 and 181). However, we did not cite the study on humans by Clark et al, 2019, as we excluded human studies from our analysis (see lines 110-112).

Line 155-158: please, report which statistics where encountered (F-tests, Spearman rank correlations...) and which equations (or where they came from) were used to calculate r? What did the authors do if they found more than one type of statistic for the same analysis? Did the authors used an a priority list to choose among them? Last, did the authors extract data from plots (e.g. using the R package metaDigitise; Pick et al. 2019), it seems so, but I did not find this information in the methods.

All studies presented estimates of relatedness rather than statistical tests. We extracted information on the mean and standard deviation of r as well as the sample size. Where these were not available in the text, we extracted data using Webplotdigitizer (Rohatgi, 2020). We have now included this step in the methods (see line 167-179) and updated our

data table (Table S1) to include all extracted summary statistics for *rPairs*, *rAverage* and *rDepression*.

Line 169-172: how more likely? I would expect this topic to particularly suffer from publication bias, particularly depending on the method used to estimate inbreeding (see comment above). Did the authors test so? (see recommendations below).

In general, inbreeding is likely to result in deleterious consequences as a result of offspring having a higher risk of being homozygous for deleterious alleles (e.g. (Charlesworth and Charlesworth 1987; Charlesworth and Willis 2009; Hedrick and Garcia-Dorado 2016)). Thus, in species where inbreeding depression had not been investigated we assumed they would suffer the deleterious consequences of mating with relatives, although we acknowledge that this is not always the case (Kokko and Ots 2006; Szulkin et al. 2013) (plus see lines 83 and 395). In addition, we examined the sensitivity of our results to this assumption by examining estimates of *ZrPairs* in species where inbreeding was ‘unclear’ (which in this version of the manuscript we have changed to ‘not mentioned’ to avoid confusion) versus those where inbreeding depression had been examined (Model S6, see Figure S9, Tables S20 and S22). This analysis showed that estimates of *ZrPairs* were similar in species where inbreeding depression was ‘not mentioned’ and where it has been confirmed suggesting this assumption generally holds. We have now added clarified this in the supplementary information (lines 50-56 plus lines 124-135 of supplementary information).

Line 185-186: I recommend calling the analyses “Bayesian phylogenetic multilevel meta-regressions” to make it clear when effect sizes were weighted by the inverse of the sampling variance. I personally had to look at the code to understand this.

Thanks for this suggestion, we now refer to the analysis as ‘Bayesian phylogenetic multilevel meta-regressions’ in the manuscript (see line 199-200).

Line 187: please, provide the version of all R packages for reproducibility purposes (Pasquier et al. 2017).

We have now added a file of all our scripts and an RData file with the data files and model objects from our analyses. Saved within this RData file is the session information so that readers have full information about the loaded packages. We have referenced the RData file on line 297 (uploaded to dryad (temporary link for review process: (uploaded to dryad: <https://datadryad.org/stash/share/MQCP6J9rPsdIxeUDJfStB-t6jn2aft2s8G2SkvEVEr0>, doi:10.5061/dryad.6hdr7sr0t).

Line 191-192: “1/(n-3)” instead?

Corrected (see line 206).

Line 194-196: as far as I can see from the code, rather than the number of individuals in the population, what is used is the number of paired individuals. Wouldn't this number generally be different to the number of individuals used to estimate *ZrAverage*?

We use the number of individuals in the population ($n_{\text{individuals}}$) rather than the number of pairs in our analysis. We have now added extra annotation to our code to aid clarity.

Line 205-206: shouldn't the wording rather be something like: "if evidence for inbreeding avoidance differ between species with and without inbreeding depression"? Also, "statistically significantly" (and throughout).

We have changed the wording of this paragraph in light of these suggestions (see lines 222-229). In addition, we have added in 'statistically' (see line 227) and throughout the rest of the manuscript (see lines 275, 329, 365, 372 and 376).

Line 215: Figure 3 indicates 39 data points rather than 38. Is that correct?

We have now updated all our figures with the updated analysis. We have checked and this figure now has the correct number of data points (34 data points as species which do not have inbreeding depression were excluded from this part of the analysis, see Figures 1-3).

Line 243-244: from the code it seems that the authors randomly chose one model of the three, so I would state that here.

Yes, this is correct. We have now stated in the manuscript that we randomly chose one model out of the three (see line 274).

Line 298-299: I'm really not sure how appropriate this number is since it is a mean of means, and therefore uncertainty of the original means is neglected. It might be better to provide meta-analytic means here.

We chose to give the untransformed values of both the mean relatedness between breeding pairs and the mean relatedness between pairs in the population. We made this decision to give untransformed values, instead of giving the transformed meta-analytic means, as we thought it would be easier to interpret and facilitate biological interpretation for the reader. In this opening paragraph of the results (lines 302-312), we are trying to give a general picture of the data distribution in our analyses and so we think that the untransformed relatedness values may be of use. For the reader interested in meta-analytical means we have provided full details of all analyses in table 2 and tables S2-S20.

Line 311: what is "PM"? Apologies if I missed it.

PM stands for posterior mean; we have now added this in (see line 304).

Line 336-339: but the effect is not that far from statistical significance, so I would interpret these results with caution (as done for some other results shown below; more on this right below).

We have edited the language in this paragraph to interpret the results with more caution (see lines 326-338).

Line 373-376: I personally prefer this interpretation (i.e. prefer to avoid hard dichotomies), but the authors would need to adjust their interpretations of small but not statistically significant p-values throughout to be consistent - in this example CI's overlap with 0 and p-value is not < 0.05.

This paragraph has now been removed from the manuscript.

Line 380: please report the estimates or refer to the table that contains them.

We have now added the estimates into the manuscript (line 355-368) and referred to the table that contains them (Table 2 & S6).

Line 393-395: I would suggest to interpret these results more cautiously since only 7 data points were available for that specific regression, and despite non-statistical significance, the regression tends to be less steep than 1:1

We have edited this statement to make our interpretation more cautious and changed our phrasing on the earlier interpretation of this results and note that this is based on 7 data points (see line 333-334).

Line 395-396: see my comments about considering this scenario when searching for studies (i.e. adding "preference" to the search string). Is Figure S2 correct here?

Done. We have also removed the reference to Figure S2.

Line 408: I would suggest "might not be effective on its own". As suggested in the next sentence, I don't think we can be sure that if a mechanism is not reported for a species, that means it does not happen.

We have now rephrased this sentence (see line 411-412) to read (edits in bold):

*'Multiple mechanisms of inbreeding avoidance were not reported in the same species, suggesting that each mechanism **might be** effective on its own.'*

Line 429: the following reference seems relevant here (and throughout): Avilés and Purcell 2012

Reference added (see line 431).

Line 451: I miss a paragraph discussing the potential limitations of the study (e.g. relatively few effect sizes, etc), what heterogeneity means for the interpretation, etc.

In lines 411 to 427 we discuss some potential limitations of the study in light of the fact that few studies look for multiple mechanisms of inbreeding avoidance. In addition, we have extended part of our discussion in response to Reviewer 2's comments (see comment 7 lines 403 to 409).

Line 461: “The data and code in this study”

Corrected (see line 465, Supplementary R file and Table S1).

Figure 1: I would recommend to add uncertainty for both axis to each dot (e.g. Figure 3 from Winney et al. 2018) so that the interpretation (e.g. line 325 and 335-336) is easier and more precise.

We made figure 1 with uncertainty added for each dot, but this made the figure very difficult to interpret. Instead, we have modified figure 2, removing regression lines and adding x and y uncertainty where data are spread across two panels and therefore not so cluttered. In figure 1 we have also added regression lines with 95% confidence intervals to give some general indication of variation in the data (See figures 1&2).

Figure 3: is the y-axis wording correct?

We have now changed this to “Probability of random mating” (see Figure 3).

Table 2: consider showing residual variance for non meta-analytic models too.

We have now remade Table 2.

Supplementary material

Minor: Olson et al. 2012 reference misses the last author, I believe.

Corrected (see line 148 of Supplementary Material).

Table S5: there seem to be duplicated rows.

Thank you for pointing this error out, some of the studies are duplicated as there are multiple species within the same study, we have now highlighted these cases in green for clarity (see Table S24). Any other duplications have now been removed.

Data

Please add the metadata provided in the supplementary material to the code and/or to an extra tab (or readme) in the excel file. Furthermore, please provide more bibliographic information (e.g. doi) in the dataset for the references included (and the excluded ones from Table S5) so that a reader can easily find them without having to match the supplementary material pdf with the dataset. All that would increase data reusability.

We have changed the format of the readme to an excel sheet which we have uploaded to dryad (uploaded to dryad: <https://datadryad.org/stash/share/MQCP6J9rPsdIxeUDJfStB-t6jn2aft2s8G2SkvEVEr0>, doi:10.5061/dryad.6hdr7sr0t) along with the supplementary code

(Table S1). We have also added in the full references to the dataset to increase reproducibility (see Table S1).

Overall, I would recommend a little bit of more data cleaning even if that will not change any results but just to increase standardization as much as possible. For example, remove unnecessary spaces in some levels (e.g. “cincta”, “reticulata”, see also variables “inbreeding_avoidance” and “Reported_in_study”), use the same format throughout (e.g. references, capital first letter for common_name), etc.

We have reformatted and cleaned the dataset (see Table S1) with a readme incorporated in the excel sheet for added clarity.

Cayuela et al., 2017: shouldn't the sample size be 50 rather than 60? Did the authors perform a data extraction double-check? If not, I would recommend to perform a double-check of a percentage of the data to confirm that data is generally correct (e.g. 25% as in Moran et al. 2020).

We manually checked the data from each of the papers when clarifying where in the manuscripts we extracted the data from. In light of the comments above, we have now removed the *Cayuela et al 2017* study from the analysis as it does not fit within our stricter inclusion criteria.

Code

The format of the dataset provided is .xlsx, whereas the code tries to import a .csv file. Please, confirm that the data provided is the correct one. I assumed so for my review, however, I got the following error for lines 47-49: “Error in `[.data.frame](data, rh.idx, ids.var, drop = FALSE) : undefined columns selected”, which seems to contradict my assumption since the column “obs_or_expt” is missing.

We have now completely re-written the code to increase reproducibility and clarity. All code and data files are now uploaded to Dryad (<https://datadryad.org/stash/share/MQCP6J9rPsdIxeUDJfStB-t6jn2aft2s8G2SkvEVEr0>, doi:10.5061/dryad.6hdr7sr0t). The scripts should run via an Rproj file that now imports the data sheet from the .xlsx file. The line of code that produced the error has been removed. The exact runs from all models are now saved in the RData object.

As a general comment, I would recommend adding more inline comments to the code to make it easier to understand it. For example, I cannot easily follow the “weighted average” performed from line 41 on.

We have now completely re-written the code and added extra annotations (available at <https://datadryad.org/stash/share/MQCP6J9rPsdIxeUDJfStB-t6jn2aft2s8G2SkvEVEr0>, doi:10.5061/dryad.6hdr7sr0t). We have also added Rstudio contents indexing so the script can be more easily navigated.

Code line 86-88: I believe this check shows that the ott_id for “simochromis pleurospilus” should be 474353 instead of 710012?

Simochromis pleurospilus is a synonym for *Pseudosimochromis babaulti* which has ott_id 710012.

Code line 301: “Method_of_IA2” does not seem to be defined beforehand in the code, and as such, “graph_1.b” cannot be recreated. Please, revise accordingly.

Done.

Model M5 does not seem to converge nor run well, particularly the random effect part of the model, but also a little bit the fixed effect part. Plus, the effective sampling is very low, actually 0 for “units”. Did the authors had similar issues?

We have now increased the run lengths, burn-ins and thinning intervals on these models. Binary models often have convergence problems, but the convergence diagnostics are now ok (potential scale reductions factors <1.1, effective sample size > 500). As these are binary models the residual variance is fixed and so the effective sampling should be 0.

Overall, I would recommend the authors to save and provide the models they ran (e.g. save them as .rda) to increase reproducibility, and also to make it easier for researchers that want to run the code without having to wait for all the models.

All models are now saved within the code file as an RData object.

Last, I feel a bit odd saying this and I might as well be totally wrong, if so, I wholeheartedly apologize, but I seem to recognize chunks of code as code I wrote for Sánchez-Tójar et al. 2020a (code: https://github.com/ASanchez-Tojar/meta-analysis_of_variance) and seemingly also Sánchez-Tójar et al. 2018 (code: https://github.com/ASanchez-Tojar/meta-analysis_sparrows_ssh)? If so, and since incentives are important in academia, I would appreciate if the authors would cite the original studies (Sánchez-Tójar et al. 2018, 2020a), the same way as we would normally do when reusing data or ideas from previous publications. If not, again, I apologize.

Our normal practice is, of course, to cite any code that is taken from other authors within the code and in the associated manuscripts and so we’re grateful to the reviewer for checking with us about this. For some reason during the code development this got lost and we can only apologise. We have now annotated the code highlighting any parts where it has been modified from other sources and cite Sánchez-Tójar et al. 2018 & 2020 (lines 219 & 294) in the main text.

Sincerely, Alfredo Sánchez-Tójar. Note that I sign all my reviews since March 2018. I am more than happy to discuss any of the suggestions I have made with the authors and/or editors (my email is alfredo.tojar@gmail.com)

Thank you, Alfredo, for all of your detailed comments, advice and providing references. We really appreciate the suggestions and hope that their inclusion has improved the manuscript.

References:

- Avilés and Purcell 2012: <https://doi.org/10.1016/B978-0-12-394288-3.00003-4>
Clark et al. 2019: <https://www.nature.com/articles/s41467-019-12283-6>
Crnokrak and Roff 1999: <https://www.nature.com/articles/6885530>
Dochtermann et al. 2019: <https://doi.org/10.1093/jhered/esz023>
Duthie and Reid 2016: <https://doi.org/10.1086/688919>
Haddaway et al. 2020: <https://t.co/QrGy3jjAuc?amp=1>
Koricheva and Kulinskaya 2019: <https://doi.org/10.1016/j.tree.2019.05.006>
Lange et al. 2017: <https://doi.org/10.1093/biolinnean/blw003>
Leedale et al. 2020: <https://doi.org/10.1073/pnas.1918726117>
Leroy 2014: <https://doi.org/10.1111/age.12178>
Moran et al. 2020: <https://doi.org/10.1111/brv.12655>
Nakagawa and Schielzeth 2010: <https://doi.org/10.1111/j.1469-185X.2010.00141.x>
Nakagawa and Schielzeth 2013: <https://doi.org/10.1111/j.2041-210x.2012.00261.x>
Nakagawa et al. 2020: <https://doi.org/10.1002/jrsm.1424>
Pasquier et al. 2017: <https://doi.org/10.1038/sdata.2017.114>
Pick et al. 2019: <https://doi.org/10.1111/2041-210X.13118>
Sánchez-Tójar et al. 2018: <https://doi.org/10.7554/eLife.37385>
Sánchez-Tójar et al. 2020a: <https://doi.org/10.1111/jeb.13661>
Sánchez-Tójar et al. 2020b: <https://doi.org/10.1111/ele.13479>
Senior et al. 2016: <https://doi.org/10.1002/ecy.1591>
Thunken et al. 2011: <https://doi.org/10.1093/beheco/arq217>
Wells et al. 2020: <https://doi.org/10.1111/ele.13578>
Winney et al. 2018: <https://doi.org/10.1111/jeb.13197>
Winternitz et al. 2016: <https://doi.org/10.1111/mec.13920>

Response to Reviewer 2

This paper addresses a topic of great general interest to biologists; that is, the evolution of inbreeding avoidance mechanisms such as mating preferences for unrelated partners. My main concern is that the paper presents a simple linear causal relationship between inbreeding depression and inbreeding avoidance, where the former selects for the latter. I completely agree that the severity of inbreeding depression is likely to be a strong selective force on inbreeding avoidance. However, the causal relationship between the two is likely to be more complex as patterns of inbreeding in the past also will determine the severity of inbreeding depression. For example, in species that have experienced frequent inbreeding in the past, there may have been purging of the rare, recessive and deleterious alleles that cause inbreeding depression. Thus, the association between inbreeding avoidance and inbreeding depression is likely to reflect a more complex causal

relationship. I don't think this makes the results any less interesting, but I suggest that the authors revise some of their predictions (see specific comment #5) and some of the interpretation of their results (see specific comment #7).

We agree that there is unlikely to be a simple causal relationship between inbreeding avoidance and depression and we have tried to add some nuance to our arguments to make sure that we don't over-simplify the interpretation as the reviewer suggests.

In response to this point, we have also worked to clarify that the main point of our analyses isn't to show an effect of inbreeding depression on avoidance but rather to show that an inbreeding avoidance isn't an inevitable consequence of depression. Selection for avoidance also depends on the risk of inbred matings occurring from random mating. It is primarily confusion on this point in the existing literature that motivated us to undertake these analyses in the first place.

We have rewritten the summary of our results in the discussion section (lines 389-393) to emphasise this point:

“Animals choose mates on the basis of relatedness when two conditions are fulfilled: when there is a risk related mates encounter each other and when there is inbreeding depression (Figure 2). If these conditions are not met, selection for mate choice based on kinship will be weak, even when inbreeding depression is extremely costly.”

Comments

(1) Lines 47–48: Please clarify what you mean by inbreeding? Inbreeding may refer to the process of mating in parental generation and/or it may refer to the production of inbred individuals in the offspring's generation. Selection for inbreeding avoidance would obviously happen at the parents' generation, whilst the fitness consequences are felt at the offspring's generation. There are also several definitions of inbreeding in the literature. For example, it may refer to mating between related individuals or non-random mating where related individuals are more likely to mate than expected by chance. For clarity, it might be useful if you state how you define inbreeding. Just to take an example where these definitions would matter: assume a small population consisting of a single male and a single female who were brother and sister. This is inbreeding according to a pedigree-based definition but not according to a definition based on random mating.

For the purposes of our analyses, we use a general definition of inbreeding to ensure that relevant studies were not excluded. We have now noted this definition in the manuscript as 'mating with relatives' (see line 49).

(2) Line 69: Is kin recognition here the same as inbreeding avoidance by mate choice? Please clarify.

Yes, here kin recognition refers to active mate choice on the basis of recognising and discriminating between related and unrelated individuals when choosing a mate was the same as inbreeding avoidance by mate choice. We have re-written this point for clarity and (see line 71-73) cited an additional study (Leedale et al. 2020).

(3) Lines 155–158: Is this your own metric for severity of inbreeding depression? If so, make this clear to the reader. If not, provide a reference for how it has been used in the past.

We have now clarified this section with an additional description of data collection (lines 167-195) and we have referenced studies using a similar metrics to estimate inbreeding depression (Crnokrak and Roff 1999; Fox and Reed 2011; Leroy 2014).

(4) Lines 158–160: Please clarify whether the focal individual that is inbred or outbred corresponds with the individual whose traits you refer to here. For example, it is unclear here whether offspring mortality and offspring mass refers to the mass of inbred offspring or the offspring of inbred parents?

The focal individual of two of the categories were the inbred individual (reproductive success and mortality) and the focal individual of the other two categories was the offspring that resulted from relatives mating so inbred offspring (offspring mortality and mass). We have clarified this in the main text (line 175-178).

(5) Lines 201–205: I'm not convinced by this argument. This relationship may also be less than one if the absence of inbreeding avoidance has led to frequent inbreeding in the past and thereby purging of rare, deleterious and recessive alleles. See also major comment.

We believe this point arises because of confusion over what we are actually testing. We think the referee has interpreted this as testing if inbreeding avoidance increases with inbreeding depression. However, these sentences only refer to how we quantified inbreeding avoidance and have nothing to do with inbreeding depression.

Specifically, we are quantifying whether mates are less related than on average for the population. This importantly allows us to test if the avoidance of related mates only occurs when relatives frequently encounter each other (average relatedness in the population is high), which is indicated by a slope less than 1. We have now tried to clarify this point in lines 227-234, but happy to look at it again if we have misunderstood the point.

(6) Lines 299–301: Please check this statement. It is unclear to me how relatedness can be negative it is bounded between 0 and 1 (see lines 130–133)?

The statement on lines 139-140 was an error that has now been corrected. It should have read that relatedness values can vary between -1 and 1. Where negative relatedness values represent individuals that are less related than average (Wang 2017). See also above comment 2 on the 'Data extraction' section of Reviewer 1's comments (p 5-6 of this document).

(7) 339–343: An alternative explanation here is that the lack of inbreeding avoidance has been associated inbreeding in the past, which has purged the population of the rare, deleterious and recessive alleles that cause inbreeding depression. See also major comment.

We have now pointed out this alternative explanation in the manuscript (see line 337-338). In addition, we have highlighted that a lack of inbreeding avoidance could be associated with past inbreeding in our discussion (see lines 405-409).

(8) Line 372: Please check this subheading and similar statements throughout. I appreciate that this is not your intention, but this might be read as a 'good for the species' argument.

This has now been removed.

(9) Lines 380–382: I'm not sure if I get this argument. Even if a given species has evolved inbreeding avoidance mechanisms, it seems unlikely that such mechanisms would be perfect. If not, there should be some risk of inbreeding and if this is the case, should we not expect inbreeding depression to be more severe when parents are more related? Is this finding robust or does it reflect low statistical power due to few instances of inbreeding between closely related parents?

This paragraph has now been removed.

(10) Lines 395–396: Consider rewording this statement. I would argue that there is always some transmission benefit (or inclusive fitness benefit) of inbreeding as it increases the probability that a given gene is transmitted. However, this benefit is often outweighed by the cost due to the reduction in fitness of inbred offspring. Presumably, this statement refers to the net benefit of inbreeding?

We have now rephrased this statement to refer to the 'net benefit of mating with relatives' (see line 395).

(11) Line 399: Individuals avoid inbreeding (not species).

Thank you for pointing out this error, '*species*' has now been changed to '*individuals*' (see line 339).

(12) Lines 439–440: Please add a reference to support this claim.

We have now supported this claim with references (see line 442-444).

Full Reference list to articles mentioned above:

- Barati A, Andrew RL, Gorrell JC, McDonald PG (2018) Extra-pair paternity is not driven by inbreeding avoidance and does not affect provisioning rates in a cooperatively breeding bird, the noisy miner (*Manorina melanocephala*). *Behavioral Ecology* 29:244–252. <https://doi.org/10.1093/beheco/ax158>
- Billing AM, Lee AM, Skjelseth S, et al (2012) Evidence of inbreeding depression but not inbreeding avoidance in a natural house sparrow population. *Molecular Ecology* 21:1487–1499. <https://doi.org/10.1111/j.1365-294X.2012.05490.x>
- Bolton JL, Winland C, Ford B, et al (2012) Kin discrimination in prepubescent and adult Long-Evans rats. *Behavioural Processes* 90:415–419. <https://doi.org/10.1016/j.beproc.2012.04.008>
- Bordogna G, Cunningham G, Fitzpatrick LJ, et al (2016) An experimental test of relatedness-based mate discrimination in a social lizard. *Behavioral Ecology and Sociobiology* 70:2139–2147. <https://doi.org/10.1007/s00265-016-2217-9>
- Brekke P, Wang J, Bennett PM, et al (2012) Postcopulatory mechanisms of inbreeding avoidance in the island endemic hihi (*Notiomystis cincta*). *Behavioral Ecology* 23:278–284. <https://doi.org/10.1093/beheco/arr183>
- Carleial R, McDonald GC, Spurgin LG, et al (2020) Temporal dynamics of competitive fertilization in social groups of red junglefowl (*Gallus gallus*) shed new light on avian sperm competition. *Philosophical Transactions of the Royal Society B: Biological Sciences* 375:20200081. <https://doi.org/10.1098/rstb.2020.0081>
- Cayuela H, Léna JP, Lengagne T, et al (2017) Relatedness predicts male mating success in a pond-breeding amphibian. *Animal Behaviour* 130:251–261. <https://doi.org/10.1016/j.anbehav.2017.05.028>
- Charlesworth D, Charlesworth B (1987) Inbreeding depression and its evolutionary consequences. *Annual review of ecology and systematics* Vol 18 18:237–268. <https://doi.org/10.1146/annurev.es.18.110187.001321>
- Charlesworth D, Willis JH (2009) Fundamental Concepts in Genetics: The genetics of inbreeding depression. *Nature Reviews Genetics* 10:. <https://doi.org/10.1038/nrg2664>
- Collet M, Amat I, Sauzet S, et al (2020) Insects and incest: Sib-mating tolerance in natural populations of a parasitoid wasp. *Molecular Ecology* 29:596–609. <https://doi.org/10.1111/mec.15340>
- Crnokrak P, Roff DA (1999) Inbreeding depression in the wild. *Heredity* 83:260–270. <https://doi.org/10.1038/sj.hdy.6885530>

- Dolotovskaya S, Roos C, Heymann EW (2020) Genetic monogamy and mate choice in a pair-living primate. *Scientific Reports* 10:20328. <https://doi.org/10.1038/s41598-020-77132-9>
- Dunn SJ, Clancey E, Waits LP, Byers JA (2012) Genetic evidence of inbreeding avoidance in pronghorn. *Journal of Zoology* 288:119–126. <https://doi.org/10.1111/j.1469-7998.2012.00932.x>
- Dunn SJ, Clancey E, Waits LP, Byers JA (2011) Inbreeding depression in pronghorn (*Antilocapra americana*) fawns. *Molecular Ecology* 20:4889–4898. <https://doi.org/10.1111/j.1365-294X.2011.05327.x>
- Duthie AB, Bocedi G, Reid JM (2016) When does female multiple mating evolve to adjust inbreeding? Effects of inbreeding depression, direct costs, mating constraints, and polyandry as a threshold trait. *Evolution* 70:1927–1943
- Fox CW, Reed DH (2011) INBREEDING DEPRESSION INCREASES WITH ENVIRONMENTAL STRESS: AN EXPERIMENTAL STUDY AND META-ANALYSIS. *Evolution* 65:246–258. <https://doi.org/10.1111/j.1558-5646.2010.01108.x>
- Grafen A (1985) A geometric view of relatedness. *Oxford surveys in evolutionary biology*
- Hedrick PW, Garcia-Dorado A (2016) Understanding Inbreeding Depression, Purging, and Genetic Rescue. *Trends in Ecology & Evolution* 31:940–952. <https://doi.org/10.1016/j.tree.2016.09.005>
- Kokko H, Ots I (2006) When not to avoid inbreeding. *Evolution* 60:467. <https://doi.org/10.1554/05-613.1>
- Leedale AE, Sharp SP, Simeoni M, et al (2018) Fine-scale genetic structure and helping decisions in a cooperatively breeding bird. *Molecular Ecology* 27:1714–1726. <https://doi.org/10.1111/mec.14553>
- Leedale AE, Simeoni M, Sharp SP, et al (2020) Cost, risk, and avoidance of inbreeding in a cooperatively breeding bird. *Proceedings of the National Academy of Sciences* 117:15724–15730. <https://doi.org/10.1073/pnas.1918726117>
- Leroy G (2014) Inbreeding depression in livestock species: review and meta-analysis. *Animal Genetics* 45:618–628. <https://doi.org/10.1111/age.12178>
- Mishra A, Tung S, Shree Sruti VR, et al (2020) Mate-finding dispersal reduces local mate limitation and sex bias in dispersal. *Journal of Animal Ecology* 89:2089–2098. <https://doi.org/10.1111/1365-2656.13278>
- Rabier R, Robert A, Lacroix F, Lesobre L (2020) Genetic assessment of a conservation breeding program of the houbara bustard (*Chlamydotis undulata undulata*) in Morocco, based on pedigree and molecular analyses. *Zoo Biology* 39:422–435. <https://doi.org/10.1002/zoo.21569>

- Robinson SP, Kennington WJ, Simmons LW (2012) Preference for related mates in the fruit fly, *Drosophila melanogaster*. *Animal Behaviour* 84:1169–1176. <https://doi.org/10.1016/j.anbehav.2012.08.020>
- Robinson SP, Kennington WJ, Simmons LW (2009) No evidence for optimal fitness at intermediate levels of inbreeding in *Drosophila melanogaster*. *Biological Journal of the Linnean Society* 98:501–510. <https://doi.org/10.1111/j.1095-8312.2009.01301.x>
- Sánchez-Tójar A, Lagisz M, Moran NP, et al (2020) The jury is still out regarding the generality of adaptive ‘transgenerational’ effects. *Ecology Letters* 23:1715–1718. <https://doi.org/10.1111/ele.13479>
- Szulkin M, Stopher KV, Pemberton JM, Reid JM (2013) Inbreeding avoidance, tolerance, or preference in animals? *Trends in Ecology & Evolution* 28:205–211. <https://doi.org/10.1016/J.TREE.2012.10.016>
- Van Dijk RE, Covas R, Doutrelant C, et al (2015) Fine-scale genetic structure reflects sex-specific dispersal strategies in a population of sociable weavers (*Philetairus socius*). *Molecular Ecology* 24:4296–4311. <https://doi.org/10.1111/mec.13308>
- Varian-Ramos CW, Webster MS (2012) Extrapair copulations reduce inbreeding for female red-backed fairy-wrens, *Malurus melanocephalus*. *Animal Behaviour* 83:857–864. <https://doi.org/10.1016/j.anbehav.2012.01.010>
- Vayssade C, de Fazio C, Quaglietti B, et al (2014) Inbreeding Depression in a Parasitoid Wasp with Single-Locus Complementary Sex Determination. *PLOS ONE* 9:e97733. <https://doi.org/10.1371/journal.pone.0097733>
- Wang J (2017) Estimating pairwise relatedness in a small sample of individuals. *Heredity* 119:302–313. <https://doi.org/10.1038/hdy.2017.52>
- Wright S (1992) Coefficients of Inbreeding and Relationship. *The American Naturalist* 56:330–338

Appendix B

Department of Zoology
University of Oxford
Zoology Research and Administration Building
11a Mansfield Road
Oxford
OX1 3SZ
Email: victoria.pike@zoo.ox.ac.uk

Dear Dr Dall and Professor Barrett,

Please find attached the resubmission of our manuscript RSPB-2021-1045 entitled “Why don’t all animals avoid inbreeding?”.

We are very grateful to the reviewers for their positive and constructive comments: over the revision process, they have made suggestions that have led to substantial improvements to our manuscript.

We have now clarified the comments that Reviewer 1 raised in reference to potentially overfitting our models and addressed all the minor comments. All our changes in response to the reviewer comments are discussed below. We have also added a new reference to a recent paper in *Nature Ecology and Evolution* (de Boer et al., 2021) which highlights the need for a general explanation as to why many species do not avoid inbreeding - this is exactly what our paper sets out to provide.

Yours sincerely,

Victoria Pike

Reviewers comments bolded, our responses normal text, line numbers refer to the manuscript without tracked changes.

Associate Editor

Comments to Author:

The authors should discuss potential problems that might occur with overfitting the models.

Thank you for this comment, please see our responses to Referee 1's comments below.

Reviewer(s)' Comments to Author:

Referee: 2

Comments to the Author(s).

The authors have address all my comments on the previous version of the manuscript and they an excellent job in revising this manuscript. This manuscript would in my opinion make a valuable contribution to the field.

Per Smiseth (I sign all my reviews)

We thank Per for his thoughtful review which helped us greatly in improving the clarity of our manuscript.

Referee: 1

Comments to the Author(s).

I have now carefully read the authors' responses to the reviewers, and the new version of the manuscript. I am very satisfied with the changes implemented. The authors have made an excellent and thorough job dealing with the concerns of the reviewers. Thank you very much. I'm convinced that the study has consequently improved. I have only a few more comments and suggestions left, mostly minor, but two potentially major ones, that I recommend dealing with.

Many thanks your continued support in helping us improve our manuscript. Please find our detailed responses to your additional comments below.

1. Some models seem to be over-fitted, so I would suggest either to not present them or to explicitly and very clearly warn the reader that their results are unlikely to be reliable. The most extreme case seems to be model 5 (line 197), which, if I understood correctly, estimates 18 fixed effect estimates plus 1-2 random effect estimates. That is, it estimates about 20 estimates, but it only includes 34 data points (is this interpretation correct?). Other models that seem to be in high risk of being over-fitted are models 3, 6, 7, and 8. As

a rule of thumb, I normally aim at 10 data points per model estimate. The minimum I have seen recommended is 4 data points per model estimate, which I personally consider extremely low. Related to this, I would recommend to remind the reader in the discussion section that unfortunately only very few studies (41 max) were available, and that we need many more studies on this topic to obtain a clearer and more precise picture about the hypotheses tested.

We are grateful to the referee for raising this issue, which is important for both interpreting our results and guiding further work. Consequently, we have edited the main text to indicate where results should be interpreted with caution due to low sample sizes, and highlighted where further data collection is needed in the discussion (see lines: 341-342, 372, 377-380, 408-424).

With respect to the number of parameters estimated in our models, there appears to be some confusion here. The number of parameters estimated in model 5 is not 18, but a single random effect and eight fixed effects (4 intercepts for each of the methods of inbreeding avoidance + 4 slopes to estimate the relationship with average relatedness). This is the equivalent of conducting a regression of *ZrPairs* vs *ZrAverage* for each method of inbreeding avoidance with the following samples size (active mate choice = 9, no avoidance = 11, sex-biased dispersal = 9, post-copulatory avoidance = 5). In models 3, 6, 7 and 8 four fixed effects are estimated.

The number of data points required to estimate parameters is a debated issue and recommendations vary. To take a few examples, Forstmeier & Schielzeth 2011 and Crawley 2013 suggests 3 data points per parameter is required, while Austin and Steyerberg 2015 use stimulations to show that an average of two data points is enough to estimate regression coefficients and their standard errors.

We do agree with the referee that less than 10 data point per parameter is low. However, given the sample sizes above and that they exceed some recommendations we believe it is better to present the analyses along with caution in the main text over interpretation.

2. An I2 of 21% (line 220) is an unusually low level of heterogeneity. I would recommend the authors to discuss what this means for their results in the discussion section (see Rutkowska et al. 2014 as example). In short, that I2 suggest that most (79%) of the differences observed in results across studies are simply explained by differences in sampling variance (i.e. sample size) between studies and only 21% are potentially due to methodological and/or biological effects. For meta-analyses in ecology and evolution, normally only ~5% of the differences in results observed across studies are explained by differences in sampling variance (Senior et al. 2016), making this result, in my opinion, remarkable and worth interpreting and discussing in detail.

This is a very good and potentially biologically interesting point. In contrast, to relatedness between pairs (*ZrPairs*), tests of heterogeneity for average relatedness in populations (*ZrAverage*) and inbreeding depression (*ZrDepression*) show high and significant heterogeneity. This suggests that the lack of heterogeneity in *ZrPairs* is not due data structure or some methodological issue. Instead, it appears there is a lack of variation in

ZrPairs, which is expected if inbreeding avoidance eliminates high values of relatedness between pairs.

We have now added additional analyses (Models 1.4 to 1.9 in R script 'Data analyses') to estimate heterogeneity in *ZrAverage* and *ZrDepression* to contrast them with *ZrPairs* (lines 206-213). We have re-written the first paragraph of the results presenting these results to highlighting that inbreeding avoidance may reduce heterogeneity in relatedness between pairs (lines 293-301).

Minor:

Line 207-210: I'm unsure about adding a predictor's SE to the model. As far as I can see, it simply tests whether the uncertainty of the predictor is related to the response variable, which I don't think is what the authors intended to do when adding this predictor? Could the authors expand on this?

Our intention was to account for differences in the accuracy with which predictors were measured. However, as the referee points out it tests whether uncertainty in the predictor relates to the response variable. The sampling variance in predictors (*ZrAverage* & *ZrDepression*) is also largely correspondent to the sampling variance in *ZrPairs* due to estimates coming from the same studies and therefore similar sample sizes. As a result, we have removed predictor SEs from the models and removed the lines 207-210 from the main text. All affected analyses we re-run and our results are unchanged.

Line 237-242: I did not know it was possible to run a binary meta-analysis in MCMCglmm. For the weights, I wonder whether it might make sense to weight by $1/(n-3)$ just to keep consistency with before, even though the response is not a Zr.

The response variable is not a correlation coefficient and therefore the sampling variance is not equal to $1/(n-3)$. We therefore choose to weight by the inverse sample size. However, a more elegant way to account for variation in sample sizes across studies is to convert the binary outcome (0,1) and the number of pairs examined for each species into a binomial response with "number of successes" and "number of failures", or in our case number of pairs that exhibited inbreeding avoidance vs those that did not. Using this approach variation in sample sizes is taken into account across studies. We also now use this approach for examining patterns of random mating (lines 246-250).

Line 286: where effect sizes weighted by the inverse of the sampling variance in the Egger's tests too? They should, in principle, be. Same for the time-lag bias test.

Effect sizes were weighted by inverse sampling variances in the time-lag bias tests, but not in the Egger's regression. We have now re-run these analyses with weightings included and found very similar results to before – no evidence of publication bias. We have now clarified our approach in the main text (lines 272-285).

Line 303-304: I would suggest adding something like “although the effect was small and uncertain” here, and to show here the heterogeneity estimates.

We have revised the entire first paragraph of the results in response to the comment about heterogeneity above (lines 293-301). We decided not to add “the effect size was small and uncertain” as a small effect size is predicted under inbreeding avoidance and so highlighting this may potentially cause confusion.

Line 315: I would recommend adding how many out of how many species to be more explicit here; e.g. “In most species where inbreeding depression has been reported (X out of X),...” . Same in line 326.

We have now changed this to include the number of species (see lines 305 and 316).

Line 450-451: I find it a bit confusing that all of the sudden female fig wasps are introduced here. I had to re-read the paragraph a couple of times.

In our study, we highlight that inbreeding avoidance isn't an inevitable consequence of inbreeding depression but selection for avoidance also depends on the probability that related males and females encounter each other. In our final paragraph we put our findings into a broader biological context where kin selection does not disfavour other deleterious behaviours because relatives rarely meet. We highlight three specific behaviours: kin discrimination, paternal care and sex-ratio adjustment. Fig wasps are introduced as an example whereby the optimal sex-ratio of depends upon the number of foundresses, which determines relatedness between competing offspring. Here the production of a non-optimal sex ratios as the number of foundresses varies is explained by the frequency with which females experience these conditions in nature (line 460) (Herre,1987). To help clarify our point, we have changed wording from the original text to the following:

“Evidence for the importance of risk on the strength of selection for optimising behaviour also comes from: 1) kin discrimination, where helping kin has a fitness advantage but actively recognising kin only occurs when related and unrelated individuals are frequently encountered (Cornwallis et al., 2009); 2) paternal care, where caring for unrelated offspring is costly, but adjustment of care by males is not always needed because cuckoldry is rare (Griffin et al., 2013), and 3) sex ratio adjustment, where producing more female offspring is beneficial for single foundresses, but this does not occur in species where multiple females typically lay together (Herre, 1987).” (See lines 439-451).

Line 466: I tried to access the data using this doi, but could not. I'm guessing the link will be active after acceptance.

The link provided in the MS will be active after acceptance.

Figure 3: the authors could present the 95% CrI rather than SE to keep consistency and make it more easily interpretable.

This has now been changed (see Figure 3 and Figure legend).

Table 2: is the pMCMC of model 2 difference from 1 correct? I recommend reporting the random effect and residual variance estimates for each model either here or in the supplements.

We have now removed Table 2 as we are very close to the page limit and all our model outputs are available in the supplementary tables. To make this clearer we have now changed this to <0.0001 in Table S3. Phylogenetic, residual and total variances for intercept only models of *ZrPairs*, *ZrAverage* and *ZrDepression* are now presented in Table S2.

Supplementary material: if possible, I would include the tables in the file “Supplementary Info.pdf”, it was a bit cumbersome to navigate the tabs and adjust column size in the excel file.

We understand that these tables may be cumbersome to navigate, but we feel this is the best format for them. We have now combined all the tables into a single excel file (rather than three files) and saved the column width so that they are easier to navigate than the previous version. Combining them into a PDF would make an extremely large document due to the number of tables (26 supplementary tables), some of which are extremely long (e.g. Table S25 has 1509 rows and 46 columns). We felt it was easier to access the supplementary data tables if they were all presented in the same format and documents rather than some as part of the supplementary info pdf and others as an excel file.

SM Line 77: “no statistically significant interaction”?

Added, see line 77 of supplementary material.

SM Line 143-144: in this case it could also be driven by the very small number of effect sizes available for the pedigree estimate ($n=6$), so I would highlight this potential cause as done in some of the analyses above.

We have now included a sentence pointing out the small number of effect sizes (see line 142-144 of supplementary material).

Edits/typos:

Line 67: “sex-biased” (line 366)

Corrected (see lines 67 and 359).

Line 242: I guess $N_{\text{species}} = 34$ could be deleted since that information is also provided in the following sentence.

This has now been removed.

Line 358: “was”?

Corrected (see line 348).

References cited in our responses

Austin PC, Steyerberg EW. 2015. The number of subjects per variable required in linear regression analyses. *Journal of Clinical Epidemiology* **68**:627–636. doi:10.1016/j.jclinepi.2014.12.014

Crawley MJ. 2013. *The R book*, Second. ed. Chichester, England ; Hoboken, N.J: Wiley.

de Boer RA, Vega-Trejo R, Kotrschal A, Fitzpatrick JL. 2021. Meta-analytic evidence that animals rarely avoid inbreeding. *Nature Ecology & Evolution*. doi:10.1038/s41559-021-01453-9

Forstmeier W, Schielzeth H. 2011. Cryptic multiple hypotheses testing in linear models: overestimated effect sizes and the winner's curse. *Behav Ecol Sociobiol* **65**:47–55. doi:10.1007/s00265-010-1038-5

Herre EA. 1987. Optimality, plasticity and selective regime in fig wasp sex ratios. *Nature* **329**:627–629. doi:10.1038/329627a0